# A comparative analysis of UV nadir-backscatter and infrared limb-emission ozone data assimilation

Rossana Dragani

European Centre for Medium-Range Weather Forecasts, Shinfield Park, RG2 9AX, Reading, UK

*Correspondence to:* Rossana Dragani (rossana.dragani@ecmwf.int)

**Abstract.** This paper presents a comparative assessment of ultra violet nadir-backscatter and infrared limb-emission ozone profile assimilation. The Meteorological Operational Satellite A (MetOp-A) Global Ozone Monitoring Experiment 2 (GOME-2) nadir and the ENVISAT Michelson Interferometer for Passive Atmospheric Sounding (MIPAS) limb profiles, generated by the ozone consortium of the European Space Agency Climate Change Initiative (ESA O3-CCI), were individually added to a

reference set of ozone observations and assimilated in the European Centre for Medium-Range Weather Forecasts (ECMWF) data assimilation system (DAS). The two sets of resulting analyses were compared with that from a control experiment, only constrained by the reference dataset, and independent, unassimilated observations.

Comparisons with independent observations show that both datasets improve the stratospheric ozone distribution. The changes inferred by the limb-based observations are more localized and, in places, more important than those implied by

the nadir profiles, albeit they have a much lower number of observations. A small degradation (up to 0.25 mg/kg for GOME-2 and 0.5 mg/kg for MIPAS in the mass mixing ratio) is found in the tropics between 20 and 30 hPa. In the lowermost tropo-sphere below its vertical coverage, the limb data is found able to modify the ozone distribution with changes as large as 60%. Comparisons of the ozone analyses with sonde data show that at those levels the assimilation of GOME-2 leads to about 1 Dobson Unit (DU) smaller root mean square error (RMSE) than that of MIPAS. However, the assimilation of MIPAS can still

improve the quality of the ozone analyses, and - with a reduction in the RMSE up to about 2 DU - outperform the control experiment thanks to its synergistic assimilation with total column ozone data within the DAS.

High vertical resolution ozone profile observations are essential to accurately monitor and forecast ozone concentrations in a DAS. This study demonstrates the potential and limitations of each dataset and instrument type, as well as the need for a balanced future availability of nadir and limb sensors, and long-term plans for limb-viewing instruments.

## 1 Introduction

Since the discovery of its global decline in early 1980s (Farman et al., 1985; Solomon et al., 1986), ozone has attracted the interest of both the scientific community and policy-makers (e.g. WMO, 2014a, b, and earlier assessments), as well as of the general public. Such an interest is driven and justified by the crucial role ozone plays in the chemistry and in the thermal structure of the atmosphere: a change in the amount of ozone could lead to a warming/cooling of the Earth (depending on the

altitude where the change occurs), it could affect the Earth's climate (e.g. McLinden and Fioletov, 2011), and human life - both

as consequence of its stratospheric decline, which would lead to increased ultraviolet radiation reaching the surface (discussed in a number of quadrennial assessments and progress reports of the United Nations Environment Programme, UNEP, e.g. UNEP, 2012), and for its role as pollutant, e.g., EPA (2015), Madronich et al. (2015).

The concern for the ozone decline - at an observed rate for the global total column of 2.5% between the 1980s and early 1990s (WMO, 2014a) - led to the signing of an international treaty (the 1997 Montreal Protocol and subsequent amendments) to regulate the release in the atmosphere of Ozone-Depleting Substances (ODSs, e.g. SPARC, 1998; Randel and Wu, 2007). The 2014 WMO (acronyms not defined in the text can be found either in Table 1, if they refer to satellite platforms and instruments, or in Table 2) assessment of ozone depletion stresses that although recent studies agree that the start of the 21$^{st}$ century represented a turning point in the global total column ozone trend, which now sees a slow increase in its abundance, it is not yet clear whether the current global increase can be attributed to a reduction in the amount of ODSs (WMO, 2014a). Projections of the ozone future evolution seem to agree that the ozone amount will recover towards the stratospheric levels registered before the 1980s by the end of the current century (e.g. Barnes et al., 2014). However, they do not agree on when such a recovery will be achieved (Velders and Daniel, 2014). Thus, the attention in closely monitoring the ozone evolution remains high.

Satellites have been critically important in delivering valuable information to continuously monitor this important atmospheric gas over the last four decades. The first satellite ozone measurements date back to the early 1970s when the nadir-pointing BUV instrument was launched on board of the Nimbus-4 satellite (Heath et al., 1973) followed by a successful number of similar instruments (SBUV and SBUV/2) on Nimbus-7 in 1978 and several NOAA platforms (Bhartia et al., 1996) that stretch up to present. In addition, a large variety of instruments to measure ozone have been launched since 1978 and many more are planned for the next ten to fifteen years. An account of the past and planned missions able to deliver ozone measurements can be found at http://www.wmo-sat.info/oscar/gapanalyses?view=108. This wealth of ozone observations offer the opportunity to derive a record of over forty years to study the ozone variability and changes, as well as derive trends. However, they are highly inhomogeneous in viewing geometry (nadir, limb, occultation, or a combination of them), in observed spectral range (UV, visible, near-infrared, thermal infrared, microwave) and spectral resolution, in spatial coverage (GEO vs. LEO platforms) and spatial resolution (ranging from a few hundred kilometres to a few kilometres).

A powerful way to integrate and exploit such a wealth of observations in a way that is consistent with their uncertainties is within a Data Assimilation System (DAS) with many successful examples covering the medium-range weather forecasting and reanalysis available in the literature (e.g. Daley, 1991; Courtier et al., 1994; Veersé and Thépaut, 1998; Rabier et al., 2000; Kalnay, 2003; Dee et al., 2011). Examples of ozone assimilation date back to the late 1990s, when NWP centres seized the opportunity of exploiting these observations to improve the radiance assimilation, and constrain the wind analyses (e.g. Derber and Wu, 1998; Hölm et al., 1999; Jackson and Saunders, 2002; Struthers et al., 2002). Polavarapu et al. (2005) presents an overview of some of the challenges of middle atmosphere data assimilation, in general, and ozone data assimilation, in particular. The latter is also reviewed by Lahoz et al. (2007). To exploit data synergy, guarantee redundancy, thus resilience to sudden changes in the observing system, and to be less sensitive to limitations of a particular instrument design, the general tendency in both NWP and reanalysis productions is to assimilate as many observation types as possible. An extensive literature exists

for both types of production. The state-of-the-art in the use of observations, particularly from satellites, in NWP was presented during the ECMWF's 2014 Annual Seminar (*Use of Satellite Observation in Numerical Weather Prediction*, 8-12 September 2014, Reading, UK, proceedings available at http://www.ecmwf.int/en/learning/workshops-and-seminars/past-workshops/). In the context of recent reanalyses, an account can be found at http://www.reanalysis.org. Furthermore, within SPARC a Re-

analysis Intercomparison Project (S-RIP, http://s-rip.ees.hokudai.ac.jp/) was started in 2012 aiming at comparing a number of recent reanalysis data sets (for various key diagnostics) to understand the causes of differences between them. Worthy of mention is a S-RIP special issue in *Atmos. Chem. Phys.* on the project assessments (Eds. Haynes, Stiller, and Lahoz), http://www.atmos-chem-phys.net/special_issue829.html. In contrast to the NWP and reanalysis general tendency of using as many observations as possible, there are also cases where precise choices on data selection are made from the outset. For

instance, in the latest NASA/GMAO reanalysis (MERRA-2), Bosilovich et al. (2015) explain that their ozone analyses were constrained using the SBUV ozone products (version 8.6 McPeters et al., 2013) until 2004 when the ozone assimilation was switched to a combination of ozone products retrieved from MLS (Froidevaux et al., 2008) and OMI (Bhartia, 2002) both flying on board the Aura platform (Schoeberl et al., 2006). The former is a limb-emission sensor providing ozone profiles retrieved from microwave measurements; the latter is a nadir backscatter instrument providing total column ozone from measurements

in the visible and ultraviolet spectral range.

An analysis of the Earth Observation capability planned for the next decade up to about 2025 shows a generally good temporal coverage from nadir-looking instruments with sensitivity to ozone, on either LEO or GEO platforms (see, for instance, http://www.wmo.int/pages/prog/sat/satellitestatus.php). Many of these instruments are or will be operated as part of operational missions thus ensuring long term data provision (for instance the forthcoming Sentinels 4 and 5). The same cannot be said

for instruments with a limb viewing geometry. These satellite instruments are rarely operational and their combined data record show significant gaps. Yet, limb sounders are very important when coming to ozone monitoring as their high vertical resolution can lead to significant differences when deriving trends. Furthermore, when designed to have a sensitivity to spectral ranges such as the infrared (like in the case of the ENVISAT MIPAS sensor, Dubock et al. (2001); Fischer et al. (2008)) or sub-millimitre spectral range (like in the case of the Aura MLS sounder), these instruments can provide measurements in

night-time conditions thus overcoming one of the most severe limitation of UV-Visible measurements. Many studies have been published in the literature to assess the impact of limb ozone data, particularly from the MIPAS and MLS instruments. The results from the first assimilation trial using the MIPAS ozone profiles within the ECMWF DAS were discussed by Dethof (2003). In their results, MIPAS assimilation was found able to improve the quality of the ozone analyses at high latitudes in the winter hemisphere, in general, and provide a better characterization of the Antarctic ozone hole, in particular. These results were

confirmed by later studies. For instance, Wargan et al. (2005) discussed the improvements in the ozone analyses, particularly in the polar night region and below the ozone maximum. Geer et al. (2006) were able to reproduce accurately the unusual event of the ozone hole split that occurred in September 2002 in the Met Office system. Using two configurations of the ECMWF DAS, Dragani (2013); Dragani et al. (2015) discussed the assimilation of MIPAS using retrievals obtained after a severe instrumental problem triggered changes in the instrument spectral specifications. Their results confirmed that the assimilation of MIPAS

ozone data could substantially improve the quality of the ECMWF ozone analyses compared to a baseline only constrained by

UV-Visible ozone data. A number of studies focused on the assimilation of the MLS ozone profiles recording improvements in the stratospheric ozone distribution (e.g. Jackson, 2007; Feng et al., 2008; Štajner et al., 2008). In particular, Štajner et al. (2008) also discussed how the assimilation of the MLS ozone profiles can be exploited synergistically with total column ozone from OMI (also on board of the NASA's Aura satellite) to improve the tropospheric ozone column analyses, in addition to

the vertical ozone distribution. This conclusion is confirmed by a recent study. Lefever et al. (2015) found that a combination of ozone retrievals from nadir-looking UV-Vis instruments and the MLS could provide better constraint on the tropospheric analyses than that provided by MLS alone.

All these studies agree that limb measurements are valuable, yet the need for these observations does not match the long term availability and plans for limb instruments. Thus, do the results from the available studies and the above considerations,

as well as the pragmatic decision taken in constructing the MERRA-2 reanalysis call for more sustained and longer term plans for limb measurements than currently available?

By presenting a comparative assessment of the impact on the ECMWF ozone analyses of assimilating either nadir or limb ozone profiles to the same ozone baseline, this study aims at addressing the above question. The present work focuses specifically on the point of view of data assimilation with potential implications for a large part of the ozone research spectrum

spanning from NWP to climate reanalysis, from air quality to stratospheric trends and variability.

The paper is structured as follows: the observations used in the present study are presented in section 2; the DAS set-up is discussed in section 3. A preliminary assessment of the data quality prior the assimilation is discussed in section 4 while the assimilation results are analysed in section 5. Concluding remarks and recommendations are summarized in section 6.

## 2   The ozone datasets

This section focuses on the two ozone products that were used to address the question raised in section 1, namely datasets retrieved from measurements of the MetOp-A GOME-2 instrument and the ENVISAT MIPAS sounder. Additional ozone information was used to constrain all assimilation experiments. This is described in section 3.

Launched on board of the EUMETSAT MetOp satellites, GOME-2 (Callies et al., 2000; Munro et al., 2006) is one of the new generation European instruments. The first two GOME-2 instruments were part of the MetOp-A (October 2006) and MetOp-B

(September 2012) payloads, respectively. A third one is scheduled to be launched on MetOp-C in 2017. This series of instruments continue the long-term monitoring of atmospheric ozone started by the ERS-2 GOME and ENVISAT SCIAMACHY instruments, whilst generally characterized by much smaller pixels than their predecessors to make their observations useful for air quality forecasting (e.g. Hao et al., 2014). Like its predecessors, GOME-2 is an optical spectrometer that measures the Earth's backscattered radiance and extraterrestrial solar irradiance, in the ultraviolet and visible part of the spectrum (240-790

nm). Its high spectral resolution (between 0.2-0.4 nm) permits to obtain an excess of 4000 spectral points from four detector channels per individual measurement. The first two channels (covering the 240-315 nm and 310-403 nm spectral regions) are important for ozone retrieval. In the period used in this study (2008), the GOME-2 instrument was characterized by an orbit

swath of 1920 km, and a typical footprint in the forward scan of 80 km (across track) by 40 km (along track), allowing for a daily global coverage.

The second dataset was retrieved from the MIPAS measurements (Fischer et al., 2008). This was a Fourier transform spectrometer launched in 2002 on a Sun synchronous polar orbit on board of ENVISAT. The instrument measured thermal emission at the atmospheric limb in the mid-infrared spectral range between 4.15 and 14.6 $\mu$m (or 680-2275 cm$^{-1}$), thus permitting ver-
tical profile retrieval of several minor atmospheric constituents. It was initially operated with a high spectral resolution of 0.025 cm$^{-1}$ reduced to 0.0625 cm$^{-1}$ in January 2005 to resume operations after instrumental problems occurred in March 2004. The reduced spectral resolution led to a proportional reduction in the measurement time from 4.5 sec to 1.8 sec that was exploited to increase the number of measured spectra in each scan in order to have a finer vertical limb grid in the UTLS region and an altitude range coverage from 6 to 70 km. The reduction in the measurement time was also exploited to improve the
horizontal resolution between two contiguous limb scan measurements. The instrument was operated until April 2012 when communication with the satellite was lost.

This study focuses on the assimilation of the GOME-2 and the MIPAS ozone profiles retrieved by the ozone consortium (O3-CCI hereafter) created as part of the ESA Climate Change Initiative (CCI, Plummer, 2009, http://cci.esa.int). It is noted that several algorithms have been developed over the years to retrieve ozone profiles from the GOME-2 and MIPAS measure-
ments besides those implemented by the O3-CCI. As only the O3-CCI datasets are used here for both instruments, explicit references to the algorithm are omitted hereafter. For simplicity, the datasets will be usually referred to with the name of their corresponding instrument unless misleading. The CCI programme was established, on one hand, in response to the GCOS call for climate-quality satellite data, and, on the other hand, to realise the full potential of the ESA global Earth Observation (EO) archive. CCI aims at providing stable, long-term, satellite-based ECV data products to support climate modellers
and researchers. Particular attention is paid in characterizing the observation uncertainty and providing comprehensive, fully traceable information on calibration and validation, long term algorithm maintenance, data curation and reprocessing.

The O3-CCI retrieval scheme for nadir ozone profiles was initially developed at RAL for the ERS-2 GOME instrument (Munro et al., 1998). These ERS-2 GOME ozone profiles retrieved with the RAL algorithm were found beneficial in improving the quality of the ERA-Interim ozone reanalysis in the middle stratosphere (Dragani, 2011). The retrieval scheme is based on
an optimal estimation algorithm, which combines measurements and an *a priori* in a way consistent with their error covariance matrices (Rodgers, 2000). It uses three sequential steps to retrieve and improve the ozone information from the Hartley (266-307 nm) and Huggins (323-335 nm) bands. These provide altogether between 5 and 6 degrees of freedom for signal (Rodgers, 2000; Keppens et al., 2015)). The *a priori* is derived from the McPeters et al. (2007) climatology while the temperature and pressure profiles are taken from the ERA-Interim reanalysis (Dee et al., 2011). An empirical correction is included in the CCI
product to address the instrument throughput degradation at the shorter UV wavelengths (Lang et al., 2009). The cloud radiative transfer is not modelled explicitly. Instead an effective Lambertian surface albedo is co-retrieved. Miles et al. (2015) warns that, in presence of clouds, this solution leads to a negative bias in retrieved ozone below the cloud top. Particular attention was paid in characterizing the various sources of uncertainty in the resulting retrieval. A detailed overview of this product is presented by Miles et al. (2015).

The O3-CCI retrieval scheme for limb ozone retrievals was jointly developed by the Institut fur Meteorologie und Klimaforschung (IMK) and the Instituto de Astrofísica de Andalucía (IAA) (von Clarmann et al., 2003, 2009). The scheme makes use of all the measurements within the 740-800 cm$^{-1}$ and 1060-1110 cm$^{-1}$ spectral ranges after filtering out cloud-contaminated spectra. It assumes local thermodynamic equilibrium conditions, normally verified in the troposphere and most of the stratosphere in the selected spectral regions (Echle et al., 2000). In this study we made use of the version fv0003 dataset. A full description can be found in Sofieva et al. (2013). An ozone profile product retrieved from the MIPAS measurements was also assimilated in the ERA-Interim reanalysis between October 2003 and March 2004. However, differences exist between that product and the one used here. The one assimilated in ERA-Interim was the near-real-time product retrieved with the operational ESA level 2 algorithm, named Optimized Retrieval Model (ORM, Ridolfi et al., 2000; Raspollini et al., 2006). It was retrieved from the measurements made before the instrumental problem of March 2004 that was overcome with a number of changes in the instrument set-up, including the spectral characteristics. The CCI product consists of reprocessed ozone data available from January 2005 onwards, thereby based on measurements using the modified set-up.

Due to differences in spectral ranges and viewing geometries, the two sets of retrievals used in the present study offered different horizontal (Figure 1) and vertical (Figure 2) coverages. The geographical distribution of the GOME-2 data (top panel of Figure 1) varies with latitude and time with the daily data count that ranges from a few tens to a few hundreds over a 2° latitudinal band. In contrast, MIPAS offers a more homogenous data coverage (bottom panel of Figure 1) albeit a much lower data count than GOME-2 with a MIPAS:GOME-2 data count ratio that ranges from 1:2 up to 1:40.

The vertical coverage and vertical resolution of the two products are schematically shown in Figure 2. Overall, the O3-CCI nadir profiles (NPO$_3$ hereafter) are provided on 19 vertical levels spanning the atmosphere from surface up to 0.01 hPa. The O3-CCI limb ozone profiles (LPO$_3$ hereafter) are also derived on a fixed vertical grid consisting of 32 vertical levels spanning the region of the atmosphere from 0.05 down to 300 hPa.

## 3  The data assimilation system

The data assimilation system used here is a low resolution version of the ECMWF Integrated Forecasting System (IFS). The IFS is a comprehensive atmospheric forecasting system that simulates the dynamics, thermodynamics and composition of the Earth's atmosphere and interacting parts of the Earth-system. It includes three components: a global spectral atmospheric model, an ocean wave model, and an ocean model that simulates the ocean circulation and sea ice.

At the time of writing, the global high resolution spectral model uses a resolution truncation of T$_{co}$1279, which corresponds to a cubic octahedral reduced Gaussian grid of about 9 km grid spacing, and 137 vertical levels spanning from surface to 0.01 hPa (corresponding to an altitude of about 80km). The data assimilation is performed using a four-dimensional variational (4D-Var) data assimilation scheme (Rabier et al., 2000) formulated in terms of increments (Courtier et al., 1994; Veersé and Thépaut, 1998) and with a 12-hour assimilation window. At the time of writing, over seventy different data sources were received and monitored daily from satellite alone. Variational quality control and first-guess checks are carried out for all assimilated data. A variational bias correction scheme (VarBC, Dee, 2005) is used to automatically detect and correct for

observation systematic biases. VarBC is formulated for all observations as linear regressions of a number of bias predictors that are observation type, and sensor dependent, as well as geographically varying. The coefficients of those regressions are part of the state vector and computed during the 4D-Var minimization, thus updated every assimilation cycle. Detailed information on the ECMWF system is available at www.ecmwf.int/en/research/modelling-and-prediction.

In this forecast model and analysis system, ozone is a prognostic variable (Dethof and Hólm, 2004). The ozone model is parameterized according to an updated version of the Cariolle and Déqué (1986) scheme (CD86 hereafter). In CD86, the ozone photochemistry is parametrized as a linear relaxation towards a photochemical equilibrium for the local value of the ozone mixing ratio, the temperature, and the overhead ozone column. In addition, an ozone destruction term that depends on the equivalent chlorine content for the actual year is included to parameterize the heterogeneous chemistry. For simplicity, the

CD86 scheme and the additional term for the heterogeneous chemistry will be referred to as *modified CD86*. The coefficients of the modified CD86 linear regression are updated regularly over the years (thanks to the collaboration with Daniel Cariolle, CERFACS, France). These are calculated as described in Cariolle and Teyssédre (2007). It is noted that the forecast model presents a number of limitations, for example the coefficients are produced with a 2D model that does not include explicitly the heterogeneous chemistry, which had to be included in the IFS via an additional term. Work is on-going to test alternatives

that could address these points. The preliminary assessment shows encouraging results in terms of the impact on the medium and long forecast ranges (B. Monge-Sanz, personal communication).

The variational bias correction scheme originally introduced to automatically detect and correct for observation systematic biases in the radiances (Auligné et al., 2007) was later extended to retrieved ozone products (Dragani, 2009). For relevance in the discussion, it is noted that accounting for the vertical sensitivity of any retrieved product as provided by the data averaging

kernels (AKs) is currently not possible in the IFS. Preliminary tests assimilating ozone retrievals with AKs were performed using a modified version of the IFS that was developed as part of the series of FP6, FP7 and H2020 funded projects GEMS (Hollingsworth et al., 2008), MACC Simmons (2010) and its follow-on projects, and here referred to, for simplicity, as the MACC-IFS. The results obtained from these tests compared to the same assimilation without AKs were at best neutral (A. Inness, personal communication), and thus pursued no further. When the AKs are neglected, retrieved observations, including

ozone data, are assimilated with a box-car approximation, in which each AK function is assumed to be 1 over the layer it refers to and 0 otherwise (i.e., with perfect AKs). By design of variational data assimilation methods, the location where an observation can be expected to have an impact on the analysis depends on both the background error and the region where the observation shows sensitivity, as expressed here by the AKs. As an approximation of what happens within the data assimilation, the largest impact can be expected in the region where their convolution is maximum. Han and McNally (2010) showed an

illustration of it applied to the assimilation of IASI radiances where the IASI Jacobians were used instead of the AKs. With a box-car approximation, the vertical spread of the ozone information provided by the assimilated ozone observations depends on the background error variances and covariance for ozone. This is particularly the case of the assimilation of $TCO_3$. An example of background error profile and vertical correlation matrix obtained from Dragani and McNally (2013) is given in Figure 3. This figure shows that the ozone background error is largest in the stratosphere between 20 and 80 hPa (panel a), implying that the ozone increments generated by the assimilation of $TCO_3$ products is most likely spread over this region of the

atmosphere. Similar considerations are less straightforward when assimilating profiles, as the region of the atmosphere where an impact can be made also depends on the observation error characteristics.

Because of the explicit mention, it is noted that differences exist between the ozone analysis system used here, and the one that was used in the ERA-Interim reanalysis discussed in detail by Dragani (2011), as well as the one that used for the MACC reanalysis (Flemming et al., 2009; Inness et al., 2013). These differences are summarized in table 3 for the period considered in this study. For other overlapping periods between the two ozone reanalyses, the reader is advised to refer to the corresponding literature.

## 3.1 Experiment set-up

Three assimilation experiments (a control, referred to as Exp/Ctrl, and two perturbation experiments) were run for the period July-October 2008 using a low resolution version of the IFS with an horizontal truncation of $T_l 511$, which corresponds to about 40 km grid resolution, and 91 vertical levels spanning the atmosphere from surface up to 0.01hPa.

With the exception of observations sensitive to ozone, the three experiments assimilated the same set of observations consisting in those available in the ECMWF archive for the period under study. Regarding the ozone observations, the Exp/Ctrl was constrained with ozone sensitive radiances in the infrared (from HIRS, IASI, and AIRS sounders as described in Dragani and McNally (2013)) and ozone retrievals. The latter consisted of a $TCO_3$ product retrieved at KNMI from the ENVISAT SCIAMACHY measurements (Brinksma, 2004; Eskes et al., 2005; Antón et al., 2011), and ozone retrievals from NOAA-16, -17 and -18 SBUV/2 instruments (Bhartia et al., 2012). These observations were all taken from the ECMWF operational data archive. It is noted that the SBUV/2 data produced by NOAA as 21-level ozone profiles were converted into a six-layer product (Top-1 hPa, 1-2 hPa, 2-4 hPa, 4-8 hPa, 8-16 hPa and 16 hPa-surface) at ECMWF.

The two perturbed experiments were then run with exactly the same configuration and observing system of Exp/Ctrl plus the assimilation of either the MetOp-A GOME-2 $NPO_3$ product or the ENVISAT MIPAS $LPO_3$ dataset. In the remainder of this paper, the former perturbed experiment will be referred to as Exp/GOME2 while the latter will be referred to as Exp/MIPAS.

## 4 Data quality analysis

A preliminary data analysis was performed for both datasets prior to the assimilation. The purpose was to determine the level of agreement between observed and modelled ozone fields, and, due to their importance in data assimilation, provide a preliminary assessment of the observation uncertainties. This is motivated by the fact that NWP-based systems tend to be conservative when using new observations. This can be achieved either by being particularly selective on the data that are actually assimilated (for instance through first-guess checks that depend on the level of discrepancy between model and observations) or by limiting their impact by inflating their provided uncertainties, which determine the weight the observations themselves have on the analyses. Although significant progress has been made in understanding and characterizing most sources of error in the observations, as well as in following best practice, limitations in those estimates still exist. This is either because some of the sources of uncertainty are particularly difficult to characterize or because they are unknown. Of

these two aspects, the former is investigated by examining the observation-minus-analysis (O-A) residuals (or simply analysis departures) that measure the discrepancy between the observations and co-located analyses. Here, the analyses were taken from Exp/Ctrl and thus not constrained by the observations under assessment. The latter aspect is instead investigated by comparing the assumed (i.e., provided) observation uncertainty with an estimate derived with the Desroziers et al. (2005) method. This

method is a simple consistency diagnostic, in which the observation error covariance matrix, $R$ can be estimated using the first-guess and analysis departures from the observations ((O-B), and (O-A), respectively), as $R = E\{(O-B)(O-A)^T\}$, where $E\{\}$ and $()^T$ indicate the expectation and the transpose operators, respectively. This diagnostic was derived in the case of an optimal analysis method and it is applicable under the assumption that the correlation scales of background and observation error are sufficiently different (Desroziers et al., 2009). This diagnostic has successfully been used to estimate $R$ with operational

observations (e.g. Bormann and Bauer, 2010; Bormann et al., 2010; Stewart et al., 2014).

Time series of the global mean analysis departures for both instruments are presented in Figure 4. To ease the comparison, these are provided in terms of integrated column over the region between 0.05 and 100 hPa, which is covered by both instruments. Despite the fact that the analyses used to derive the statistics are largely constrained by UV-retrieved ozone measurements, GOME-2 is the instrument that shows the largest mean discrepancy, with O-A differences being positive during the

four month period assessed and a mean value of about 5DU (about 1.5% of the global mean ozone column over that region, estimated as 90% of a global TCO$_3$ value of 350DU). The reason for this is still under investigation at the time of writing. Possible reasons for this outcome could be: *i.* differences in the information provided by the UV-based retrievals (i.e. the GOME-2 NPO$_3$ and those assimilated in Exp/Ctrl, the SBUV partial columns and the SCIAMACHY TCO$_3$); *ii.* differences in the information provided by the GOME-2 NPO$_3$ and the IR/O3 radiances; *iii.* the data assimilation itself (that is the model background

and/or the way the ozone observations are treated within the DAS). Without assessing the impact of each of these elements individually, conclusions cannot be drawn. Regarding point *i.*, Chiou et al. (2014) compared total column products generated from the a newer, reprocessed version of the SBUV dataset (v8.6), and the GOME-2 NPO$_3$ retrievals finding differences in the monthly zonal mean well within 1%. The MIPAS measurements indicate that during July-August the global mean ozone analyses are about 5DU too high (they were 10DU too low based on the GOME-2 data). Here, the discrepancy between the two

instruments is very likely related to different coverage of the two instruments, particularly over the high latitudes in the SH, as shown in Figure 1. During September-October, the O-A residuals for the two instruments are more similar and they both indicate an underestimation of the ozone analyses of about 5DU above 100hPa. The first-guess check implemented in the IFS discards all observations that, after they successfully pass the data quality control, show discrepancies from the background of 30DU or more over the column. Figure 4 shows that on average the observations from both instruments are well within such a

threshold, although it is noted that individual observations might have shown residuals from the background larger than 30DU.

The second aspect, which has a bearing in the level of impact of the two instruments on the ozone analyses, is the characterization of the observation uncertainty. An estimate for both instruments was derived using first-guess and analysis residuals from the observations as discussed in Desroziers et al. (2005). The left panel of figure 5 shows the relative difference between the estimated and provided uncertainty for the GOME-2 NPO$_3$ product. The uncertainty provided appears to be larger than

that estimated at all latitudinal bands from the surface up to about 5hPa while in the upper stratosphere it is smaller than the

estimated uncertainty, especially at high latitudes in the summer hemisphere (the Northern Hemisphere, NH, in this case) and at midlatitudes in the winter hemisphere (i.e. the Southern Hemisphere, SH, in this case). Despite the differences between provided and estimated uncertainties appearing to be rather large, they only represent up to about 4% of the observation values (right panel of figure 5). Keppens et al. (2015) also found an overestimation of a few percent in the comparisons of this product against ground based measurements (refer to their RAL product v2.1 in VMR). From a data assimilation perspective, an uncertainty overestimation, here seen at most levels from surface up to about 5hPa, is generally desirable to account for the effect of vertical correlations in the observation error that are normally neglected in most data assimilation systems, including the IFS. The only consequence of such an overestimation would be to limit the impact of the corresponding data, thus no correction is required in this region of the atmosphere. The underestimation of the observation uncertainty in the upper stratosphere is more of a concern. This is because underestimated uncertainties increase the impact of the corresponding observations and if that is associated with poor quality data the assimilation can lead to negative impact on the analyses and forecasts. As the ozone abundance above 5hPa substantially reduces with height, it is expected that if there was a detrimental effect on the ozone analyses as a consequence of assimilating these observations without any correction on their uncertainty, this should generally be negligible or at worst small. Thus, it was decided not to apply any correction to the GOME-2 observation uncertainties above 5hPa.

The vertical cross-section of the relative difference between the estimated and the provided uncertainty for the MIPAS LPO$_3$ is shown in Figure 6. In the lower and middle stratosphere, the MIPAS uncertainty appears to be generally overestimated compared with its estimated equivalent, with the exception of the high latitudes in the winter hemisphere (SH). The overestimation is typically within 4% relative to the observation values. This appears consistent with the residual variability estimate provided by Laeng et al. (2014) when assessing the MIPAS error budget using comparisons with several reference satellite datasets, for instance MLS. The diagnostic they use should be consistent with the one applied here for small differences in the provided and estimated uncertainties. The provided field is on average smaller than the estimated uncertainty in the upper troposphere, especially in the winter hemisphere, and in the upper stratosphere. The underestimation of the MIPAS uncertainty in the upper troposphere and in the upper stratosphere can potentially produce a negative impact on the analyses and forecasts. Using the same argument discussed above, any detrimental effect in the upper stratosphere is expected to be generally from negligible to small. Thus, no correction is made at those levels. The possible underestimation of the MIPAS uncertainty in the upper troposphere mostly affects the mid and high latitudes in the SH. It is noted that the period under consideration is July-October, thus winter and spring times in the SH. A data assimilation system strongly constrained by UV observations can show limitations in polar night conditions, while facing an important transition in springtime, when the UV data availability at these latitudes slowly restarts (as shown for instance in the top panel of Figure 1). Thus, it could be argued that the model-based estimate of the MIPAS uncertainty at high latitudes in the SH is less reliable than that provided at other latitudinal bands. A hint of a small underestimation of the uncertainty compared to its model equivalent in the middle troposphere in the southern midlatitudes can also be found for GOME-2 (Figure 5). Arguably, this could also point to problems in the assumed background errors in this region of the atmosphere and time of the year. Based on these considerations, it was decided to test the assimilation of both datasets without applying any correction on the uncertainties provided.

## 5    Assimilation results and discussion

### 5.1    Impact on the analyses

The comparison of the Exp/GOME2 and Exp/MIPAS analyses with those from Exp/Ctrl is shown in Figure 7. Both instruments contribute to an increase in the extra-tropics total column ozone, although with different amounts. In contrast, their impact differs in the tropical region. Here, the GOME-2 assimilation contributes to an overall reduction of up to 10 DU in the mean $TCO_3$ (top panel of Figure 7) while that of MIPAS shows a small increase of up to about 2 DU (bottom panel of Figure 7).

The mean vertical cross-section of the ozone analysis differences shown in Figure 8 also depicts a consistent picture between
the two datasets at most levels and latitudes, particularly in the extra-tropics. To account for the difference in the order of magnitude of the analysis residuals in the troposphere and stratosphere, the vertical cross-sections were also scaled to a mean ozone profile computed from the Exp/Ctrl ozone analyses over the three month period under consideration and shown in the bottom panels of Figure 8. On average, both datasets tend to increase the ozone analyses in a thick layer between 20 and 70 hPa with differences of up to 2 mg/kg from the Exp/Ctrl. Over this layer, the largest difference in terms of impact between the
two instruments can be seen at high latitudes in both hemispheres where the assimilation of the GOME-2 dataset seems to add more ozone with respect to the Exp/Ctrl than done by the assimilation of the MIPAS profiles. With a mean difference of about 1 mg/kg over the 20-70 hPa layer, the estimated mean column change produced by either dataset in the middle stratosphere adds up to an increase of around 24 DU.

In the upper stratosphere and in the UTLS region, the assimilation of either the GOME-2 or the MIPAS ozone profiles leads
to a reduction of the ozone analyses of an amount that depends on the instrument and latitudinal band. The largest difference is noticeable in the tropics, in the region of the ozone mixing ratio maximum around 15 hPa. This could also explain the discrepancy found in Figure 7 in the tropics.

In the lower and middle troposphere, both the assimilation of MIPAS and that of GOME-2 profiles tend to remove ozone in the extra-tropics. This is also the case for the tropical troposphere in the case of MIPAS while the assimilation of GOME-2 leads
to a tropospheric ozone increase in this latitudinal band.

An aspect that emerges from Figure 8 (bottom panels) is that the assimilation of the MIPAS ozone profiles can substantially modify the ozone distribution in the lower and middle troposphere with changes as large as 60%. Such a change occurs despite the fact that a limb instrument vertical coverage does not normally extend to pressure levels below about 300 hPa, and the observation uncertainty in the upper troposphere is normally larger than in the stratosphere, thus having a lower weight in the
data assimilation system. Yet, the structure of the changes are consistent in sign and often in amplitude with those inferred by the GOME-2 nadir ozone profiles. This aspect will be further discussed in section 5.2.2.

### 5.2    Comparisons with independent data

To determine whether the changes introduced by the assimilation of either the GOME-2 or the MIPAS ozone profiles represented an improvement or a degradation, the quality of the ozone analyses from the experiments described above was assessed in terms of the analysis agreement with independent, unassimilated ozone observations. Ozone profiles retrieved from MLS

(version 3.3) and ozone sondes available at the World Ozone and Ultraviolet Radiation Data Centre (WOUDC) were used as
independent ozone references for the stratosphere and troposphere/lower stratosphere, respectively.

The MLS observations offers a near global horizontal coverage from about 82°S to 82°N. The vertical range used spans from
about 316 hPa up to 0.01 hPa, which also coincides with the model top, and a vertical resolution of about 3 km. Froidevaux et al.
(2008) compared the previous v2.2 MLS ozone retrievals with matching ozone data from ground-based and satellite observations, and found that the differences are generally within 5% in most of the stratosphere, although residuals of 10-20% were
found in the lower stratosphere. Livesey et al. (2013) reports differences between the v3.3 and v2.2 MLS ozone profiles typically within 1-2% in the stratosphere and lower mesosphere.

The ground-based stations that are included in the WOUDC archive are spatially inhomogeneous, with the highest availability over Europe and North America. The archive includes measurements performed with both electrochemical concentration
cell (ECC), and Brewer Mast (BM) sonde types. The ECC precision was estimated by Komhyr et al. (1995) to be within ±5%
in the vertical range between 200 and 10 hPa, between -14% and +6% above 10hPa and between -7% and +17% below 200
hPa. The same order of precision was found by Steinbrecht et al. (1998) for the BM sondes. A data check is used to discard all
soundings with a pressure burst either at or below 40 hPa.

### 5.2.1 Methodology

The comparisons were performed in two stages. First, the 3D ozone analysis closest in time to the independent observation was
interpolated at the observation location. This gives a temporal mismatch of up to 3 hours between the observation sensing time
and the analysis valid time. The second stage takes care of the vertical interpolation. This is done by interpolating the profile
with the highest vertical resolution to the coarsest grid. Only the levels spanning the region of the atmosphere encompassed by
both datasets are used, and in the specific case of the ozone sondes only sounding that reached at least 40 hPa were included. In
the comparisons with MLS, the coarsest grid is represented by that of the MLS data with its vertical resolution of about 3 km
while in the comparisons with the ozone sondes the model has the lowest vertical resolution. The comparisons with MLS are
displayed as the vertical cross-section of the change in the analysis fit to the observations due to the addition of either GOME-2
or MIPAS to the reference observing system assimilated in the control experiment. Such a change, $\Delta$, is defined as follows:

$$\Delta = \left| \mathrm{STAT}\left( \mathrm{MLS} - \mathrm{Analyses}^{(\mathrm{Exp/PERT})} \right) \right| - \left| \mathrm{STAT}\left( \mathrm{MLS} - \mathrm{Analyses}^{(\mathrm{Exp/CTRL})} \right) \right| \tag{1}$$

where $\mathrm{STAT}()$ can be either the mean or the standard deviation. For either statistics, a negative value of $\Delta$ means that the
analyses from a given perturbed experiment, generically labelled above as Exp/PERT, fits MLS observations better than those
from Exp/Ctrl, thus leading to an improvement. In contrast, a positive value of $\Delta$ is associated to a degradation in the ozone
analyses.

The comparisons with the ozone sondes are instead shown in terms of mean RMS residuals, RMSE, between the sonde
profiles and the co-located analyses from the three experiments (Exp/Ctrl, Exp/GOME2, Exp/MIPAS). Thus the smaller is the
RMSE, the better is the analysis fit to the sonde measurements. For plotting purposes, the RMSE are computed and displayed

in terms of integrated column quantities. The level of agreement between the three sets of ozone analyses and the ozone sondes is also discussed in terms of a Mean Common Area score Factor (MCAF). The Common Area score Factor (CAF, e.g. as proposed by Dragani et al., 2015) is an indicator of how well the analysis profile fits the observation profile. It is defined as

$$CAF = \frac{\int_p min([\mathbf{O_3}^{(observation)}(p), \mathbf{O_3}^{(model)}(p)]) dp}{\int_p max([\mathbf{O_3}^{(observation)}(p), \mathbf{O_3}^{(model)}(p)]) dp} \tag{2}$$

where $\mathbf{O_3}$ is the ozone vertical profile at the observation location. The integrals, approximated with sums in the calculations,
are computed over the observation and model common vertical pressure grid. The score varies between zero and one with a CAF value of one implying a perfect fit to the observations. The MCAF score is based on a similar idea as the CAF, but averaged over a number of profiles and stations, for instance here it is computed for all profiles available over given latitudinal bands (the high latitudes in the SH and NH; the midlatitudes in the SH and NH; the tropics). It is important to bear in mind that, due to the sonde vertical coverage, the CAF and MCAF scores are most representative of the part of the atmosphere spanning
from the surface up to about 10 hPa at most.

In all comparisons, the ozone analyses were spatially co-located with the independent observations allowing a maximum of three hour time lag. To avoid any spin-up effect, only the results computed over the August-October 2008 period are presented.

### 5.2.2  Analysis and discussion

Figure 9 gives an indication of the improvement in the fit of the Exp/GOME2 and Exp/MIPAS analyses to MLS compared
to that of the Exp/Ctrl analyses, based on equation 1 and presented in terms of both the mean (top panels) and the standard deviation (bottom panels). Despite the large difference in data counts, the two datasets produce similar large scale structures in the analysis fit to MLS at all latitudes for most levels in the middle to lower stratosphere in both the mean (top panels) and the standard deviation (bottom panels). Comparison of Figures 9 and 8 shows that both instruments generally lead to stratospheric improvements in the analysis agreement to MLS at most latitudes, particularly in the atmospheric layer between 10 and 70
hPa, with a reduction in the mean residuals from MLS of up to about 2 mg/kg (about 40%) in the middle stratosphere. In the tropical region, both instruments lead to a slightly degraded fit (up to about 0.5 mg/kg, about 15%) of the ozone analyses to MLS between 20 and 30 hPa. Such a degradation, which is more pronounced in the case of MIPAS than in that of GOME-2, is balanced by an improved fit in the tropical regions above and below the 20-30 hPa layer. With changes up to 1 mg/kg, the improvement in the tropics outside the 20-30 hPa layer is more important for the nadir instrument than for the limb. At high
latitudes, the nadir instrument triggers a reduction in the level of agreement with MLS. Here, the changes are up to about 0.5 mg/kg (15%) in the NH (summer hemisphere) and about double (35%) in the SH (winter hemisphere). Negligible to small changes (typically less than 0.25 mg/kg, i.e. within 5%) are triggered in the upper stratosphere with a sign that depends on the latitudinal band. Overall, panels **a)** and **b)** of Figure 9 highlight a complementarity of the two observing systems in impacting the corresponding stratospheric ozone analyses. The bottom panels of Figure 9 show that the assimilation of either instrument leads to a reduction in the standard deviation in the extra-tropical middle stratosphere, although when assimilating MIPAS the impact is slightly larger (thus better) at high latitudes than in the case of GOME-2 assimilation. A small degradation can

be found in the standard deviations when GOME-2 is assimilated between 20 and 40 hPa at latitudes south of 60°S - where the fit to MLS was degraded also in the mean -, as opposed to a substantial improvement determined by the assimilation of

MIPAS. This increased noise given by the Nadir, UV-based observations is an indication of the difficulty of this observation type to provide a strong constraint to the ozone analyses and forecasts at high latitudes in the SH during winter-spring time. Furthermore, in contrast to what is found in the mean changes, here the vertical coverage of the impacted region is lower for MIPAS than that for GOME-2. Those differences can, arguably, be related to the instrument vertical sensitivity associated with the different viewing geometry. Averaging kernels that sharply peak at the level of maximum sensitivity for the limb sensor

(e.g. Ceccherini and Ridolfi, 2002) as opposed to broad and highly overlapping AKs for GOME-2 (Miles et al., 2015) allow the former to generate much more localized changes than the latter can do.

The conclusions drawn by assessing the change in the level of stratospheric agreement between the analyses and the MLS profiles implied by the assimilation of either the GOME-2 or MIPAS ozone profiles are confirmed by the comparisons with ozone sondes. The MCAF scores are computed over five latitudinal bands and divided into two contributions for the strato-

sphere and troposphere. For simplicity, the 100 hPa pressure level has been used as the separation level for the two contributions. The stratospheric and tropospheric scores for the midlatitudes in the NH, characterized by the largest availability of sonde profiles, are presented in Figures 10 and 11, respectively. The results at the other four latitudinal bands generally confirm those at midlatitudes in the NH. Overall, both the MIPAS and GOME-2 ozone profiles produce stratospheric analyses that are in better vertical agreement with the ozone sonde profiles than those from Exp/Ctrl, with the Exp/MIPAS score being slightly higher

than that of Exp/GOME2 (Figure 10). The MCAF scores are generally 20% lower in the lowermost part of the atmosphere up to 100 hPa (Figure 11) than they are in the region above (Figure 10). Although, both instruments lead to an improved agreement of the corresponding analyses to sonde profiles compared to the Exp/Ctrl, the differences with the Exp/Ctrl scores reduce significantly. In particular, both GOME-2 and MIPAS appear to produce similar levels of agreement, especially in the tropics (not shown). In the extra-tropics, Exp/GOME2 is marginally better in the NH (summer hemisphere) while Exp/MIPAS

is slightly better in the SH (winter hemisphere, not shown).

Why are the scores of the two Exp/PERT analyses so close in the troposphere? Based on Figure 2, is good vertical coverage with low vertical resolution (like for GOME-2) equivalent to having a poor vertical coverage but with high vertical resolution where available (like for MIPAS) in the troposphere? If so, what happens in the lowermost troposphere, in particularly below the level of MIPAS availability of roughly 400 hPa?

An answer to these additional questions can be found by analysing the vertically resolved fit of the three sets of analyses to the ozone sondes (Figure 12). As explained in section 5.2.1, this is expressed in terms of the RMS difference between the sonde profiles and co-located analyses, thus the smaller such a difference is, the higher the level of agreement between the analysed and observed ozone profiles becomes. Figure 12 in general confirms the results over the stratosphere discussed above, including the difficulty of the limb data in the tropics at pressure levels within the 20-30 hPa layer shown in panel **b)** of Figure 9. In the upper troposphere (down to about 400 hPa), Exp/MIPAS still shows a higher level of agreement to the sondes than Exp/GOME2. In the lower troposphere (below 400 hPa), the Exp/GOME2 assimilation generally outperforms that of Exp/MIPAS. However, in this region the assimilation of MIPAS $LPO_3$ still improves the ozone analyses compared to the control

experiment, despite a vertical coverage for the limb sounder only down to about 400hPa. Two possible mechanisms could explain the improvements in the lower troposphere: 1) a synergistic assimilation of limb ozone profiles and total ozone column data, and 2) the vertical transport. In the current system, ozone is used as a univariate variable, which means it does not affect the rest of the system in general, and the winds in particular. Moreover, with typical vertical velocity in the upper troposphere of typically 2-3 hPa/hour, the vertical transport could explain less than 50hPa vertical displacement of the information inferred by the assimilation of MIPAS data within a 12-hour assimilation window. With a vertical coverage provided by the MIPAS instrument down to about 400 hPa at best, the vertical transport alone cannot explain the changes in the lower troposphere. Based on these considerations, this positive result can only be a consequence of exploiting the synergy between the LPO$_3$ and the total column product that was assimilated in all experiments. Figure 3 in this paper (courtesy of Dragani and McNally, 2013) shows that the ozone background error variances are largest in the stratosphere between 20 and 80 hPa, implying that the ozone increments generated by the assimilation of total column products alone would most likely be spread over this region of the atmosphere. However, in this case, the combination of improved stratospheric ozone concentration obtained thanks to the assimilation of the LPO$_3$ observations, and the synergistic assimilation of the latter with TCO$_3$ products provides an indirect constraint on the ozone analyses at levels below the limb vertical coverage.

These results would suggest that the vertical resolution does matter particularly in a region like the UTLS. The retrieval vertical coverage, although important, matters to a less extent as an impact can be made in unobserved regions if the synergy with other observations can be exploited within the data assimilation system.

## 6    Concluding remarks and recommendations

The present study aims at providing a comparative analysis of UV nadir-backscatter and infrared limb-emission ozone profile data assimilation, and thus draw conclusions on how the viewing geometry impacts the ability of these two classes of sensors in constraining and improving the quality of the resulting analyses. The MERRA-2 reanalysis already represents an example where a pragmatic choice was made from the outset in whether to assimilate limb ozone profiles in place of UV nadir-backscatter ozone profiles, despite the tendency typical in NWP and reanalysis of using as many observations as possible. This consideration called for a more quantitative assessment of the relative impact on and ability to improve the quality of the resulting ozone analyses of the nadir and limb ozone observations. An analysis of the Earth Observation capability planned for the next ten to fifteen years shows a generally good temporal coverage from nadir-looking instruments measuring the backscattered solar radiation in the UV(-Vis) spectral range. Many of these instruments are or will be operated as part of operational missions thus ensuring long term data provision. In contrast, over the same time-frame the availability of measurements from limb-viewing sensors is, at best, poor, and normally confined to research satellites with limited long-term continuation, if any at all. Thus, the results are of primary interest to space agencies as input in shaping future plans for Earth Observation, and have implications for a large part of the ozone research spectrum spanning from Numerical Weather Prediction (NWP) to climate reanalysis, from air quality to stratospheric trends and variability (relevant, among others, for the activities performed within the SPARC project, http://www.sparc-climate.org/, and WMO ozone assessments, e.g. WMO, 2014a).

The main findings from this study are:

1. On average, the GOME-2 and MIPAS datasets produce changes in the vertical distribution of the ozone analyses with similar large-scale patterns, in particular an increase of the ozone analyses up to 2 mg/kg compared to the control experiment at levels between 20 and 70 hPa and an ozone amount reduction in the extra-tropical lower and middle troposphere. Comparisons with independent observations indicate that those changes improve the agreement between the resulting ozone analyses and independent observations (see points 3 to 6 below).

2. In the middle stratosphere, the changes induced by the assimilation of the MIPAS limb ozone profiles are more localized than those implied by the assimilation of the GOME-2 nadir profiles.

3. Differences in the impact were noticeable, especially in the tropics and at high latitudes. In the tropics, both instruments lead to a degraded fit to MLS (up to about 0.5 mg/kg increase in the residuals) in the layer between 20 and 30 hPa, and to an improved fit (up to about 1 mg/kg reduction in the residuals) above and below that layer. A reduction in the level of agreement between the ozone analyses and the MLS observations is also triggered by the GOME-2 assimilation at high latitudes where the mean residuals are increased of up to about 0.5 mg/kg in the NH (summer hemisphere) and about 1 mg/kg in the SH (winter hemisphere).

4. The MCAF scores show that both datasets lead to improved ozone analyses compared to the control experiment (about 20% higher MCAF score in the stratosphere and about 10% higher MCAF in the troposphere than the MCAF from Exp/Ctrl). But while the assimilation of MIPAS is better than that of GOME-2 in the stratosphere, the scores in the troposphere (here simply referred to as the layer below 100 hPa) are very similar to each other despite the fact that the limb vertical coverage does not normally extend below about 400 hPa

5. In the upper troposphere (down to about 400 hPa), the assimilation of MIPAS profiles outperforms that of GOME-2 based on the fit to the ozone sondes. The former shows RMSE values between 1 and 2 DU lower than those derived for the latter.

6. In the lowermost troposphere(below 400 hPa), the assimilation of the MIPAS ozone profiles was found able to substantially modify the ozone distribution with changes as large as 60%. Comparisons with ozone sonde profiles show that, although not always better than that of GOME-2, the assimilation of the MIPAS ozone profiles improves the fit to the independent data by reducing the RMSE of up to 2 DU compared to that of the Exp/Ctrl analyses.

The results presented in this paper highlight the complementarity of the two observing systems and confirmed how the instrument characteristics (observed spectral range and viewing geometry) shape the observation ability in constraining the ozone analyses. Overall, both types of observations can improve the vertical distribution of the stratospheric ozone analyses that are too high above 20 hPa and too low below 25-30 hPa in the control experiment. The assimilation of the nadir observations proves to be more successful than that of the limb observations in the tropical stratosphere. In contrast, the assimilation of the limb ozone profile is essential at high latitudes and in the upper troposphere. Only small differences between the two

5   perturbation experiments were seen in the lowermost troposphere that is not sampled by limb sensors but where nadir instruments, like GOME-2, can provide useful information (Miles et al., 2015). On one hand the limb limited sensitivity can only be realized here if their synergy with other ozone observations (in particular total column ozone products) can be exploited within the data assimilation system; on the other hand such a synergy cannot completely replace the improvement produced by the assimilation of nadir-based ozone profiles.

10   To accurately monitor and forecast ozone concentrations in a data assimilation system, high vertical resolution ozone profile observations are essential. With only total column ozone measurements, analyses would depend too much on the model background and the background error covariances with analysis increments that would be most likely located in the region of the atmosphere where the background error is largest.

Limb observations are essential to bring the improvements in the ozone analyses necessary for example to assess long-term stratospheric changes from reanalysis productions, especially at high latitudes, but also in the upper troposphere and across the

tropopause. However, in the lowermost part of the atmosphere important for air quality, nadir-based profiles are still the most important source of information.

The present study has demonstrated the potential that each dataset and instrument type has in its own right. The findings discussed in this study would indicate the need for a more balanced availability between nadir and limb sensors than currently exists thus supporting revised EO plans to improve the availability and long-term continuation of limb viewing instruments for

the next decade and beyond.

*Acknowledgements.* The author was supported by the European Space Agency through the Climate Change Initiative / Climate Modelling User Group (CCI/CMUG) project. A special thank-you goes to Dr. Stephen English (ECMWF) and Dr. Roger Saunders (The Met Office) for comments and useful discussion, and the three ACP reviewers and Editor for their comments and suggestions. Anabel Bowen and Simon

Witter (ECMWF) skilfully improved the quality of the figures. The ESA-funded Ozone-CCI consortium is thanked for the provision of the CCI GOME-2 and MIPAS ozone profiles. The MLS ozone data used in this study for the independent validation were the reprocessed (version 3.3) ozone profiles retrieved from the NASA Distributed Active Archive Center at the Goddard Space Flight Center. The ozone sondes were retrieved from the World Ozone and Ultraviolet Radiation Data Centre (WOUDC).

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

**Table 1.** List of acronyms and abbreviations for the satellite platforms and instruments used in this paper.

| Acronym | Definition |
| --- | --- |
| AIRS | Advanced InfraRed Sounder |
| ENVISAT | ENVIronmental SATellite |
| ERS-2 | European Remote Sensing 2 |
| GOME-2 | Global Ozone Monitoring Experiment 2 |
| HIRS | High-resolution Infrared Radiation Sounder |
| IASI | Infrared Atmospheric Sounding Interferometer |
| MetOp | Meteorological Operational Satellite |
| MIPAS | Michelson Interferometer for Passive Atmospheric Sounding |
| MLS | Microwave Limb Sounder |
| OMI | Ozone Monitoring Instrument |
| SBUV | Solar Backscatter Ultra Violet |
| SCIAMACHY | Scanning Imaging Absorption Spectrometer for Atmospheric Cartography |
| TOMS | Total Ozone Mapping Spectrometer |
| UARS | Upper Atmosphere Research Satellite |

**Table 2.** List of all other acronyms and abbreviations used in this paper.

| Acronym | Definition |
| --- | --- |
| BUV | Backscatter Ultra Violet |
| CTM | Chemistry Transport Model |
| ECV | Essential Climate Variable |
| EOS | Earth Observing System |
| ESA | European Space Agency |
| EUMETSAT | European Organisation for the Exploitation of Meteorological Satellites |
| FP6 | The Sixth Framework Programme |
| FP7 | The Seventh Framework Programme |
| GCOS | Global Climate Observing System |
| GEO | Geostationary |
| GMAO | Global Modeling and Assimilation Office |
| GEMS | Global and regional Earth-system (Atmosphere) Monitoring using Satellite and in-situ data |
| H2020 | Horizon 2020 Programme |
| IR | InfraRed |
| LEO | Low Earth Orbit |
| MACC | Monitoring Atmospheric Composition and Climate |
| MERRA | Modern Era Retrospective-analysis for Research and Applications |
| MOZART | Model for OZone And Related chemical Tracers |
| NASA | National Aeronautics and Space Administration |
| NOAA | National Oceanic and Atmospheric Administration |
| NWP | Numerical Weather Prediction |
| RAL | Rutherford Appleton Laboratory |
| SPARC | Stratospheric Processes and Their Role in Climate |
| $TCO_3$ | Total Column Ozone |
| UTLS | Upper Troposphere and Lower Stratosphere |
| UV | Ultra Violet |
| UV-Vis | Ultra Violet and Visible |
| VMR | Volume Mixing Ratio |
| WOUDC | World Ozone and Ultraviolet Radiation Data Centre |

**Table 3.** Major differences between the ozone analysis system used in this paper and those of the ERA-Interim and the MACC reanalyses. Detailed information on the set-up of these two reanalyses can be found in the corresponding literature. For the ozone-specific differences, an indication of the region of the atmosphere where an impact is expected is indicated in the fifth column. There, the **ST** and **TR** letters stand for Stratosphere and Troposphere, respectively. [+]The near real time (NRT), coarse vertical resolution profiles retrieved from NOAA-16, -17, and -18 SBUV/2 measurements. [$]Reprocessed, coarse vertical resolution profiles retrieved from NOAA-16, -17, and -18 SBUV/2 measurements. [*]NRT $TCO_3$ from the SCIAMACHY measurements made in nadir-viewing geometry. [§]Infrared radiances from AIRS, IASI, and HIRS. [†]NRT $TCO_3$ from OMI. [#]Reprocessed OMI $TCO_3$. [‡]NRT profiles from MLS (v2.2), used down to 68hPa. [¶]Reprocessed profiles from MLS (v2.2), used down to 215hPa. [††]IASI channel 1088. [‡‡]AIRS channel 1585.

| | Exp/Ctrl | ERA-Interim | MACC | Region of impact |
|---|---|---|---|---|
| **Model cycle** | CY40R1 (2013) | CY31R2 (2006) | CY36R4 (2010) | / |
| **Hor. resolution** | $T_l$511 ( 40 km) | $T_l$255 ( 80 km) | $T_l$255 ( 80 km) | / |
| **Ver. resolution** | L91 | L60 | L60 | / |
| **Top of Atm.** | 0.01 hPa | 0.1 hPa | 0.1 hPa | / |
| **X used $TCO_3$** | SCIAMACHY[*] | SCIAMACHY[*], OMI[†] | SCIAMACHY[*], OMI[#] | ST |
| **X used profiles** | SBUV/2[+] | SBUV/2[+], MLS[‡] | SBUV/2[$], MLS[¶] | ST + TR |
| **X used radiances** | IR/O3[§] | N/A | N/A | UTLS |
| **Bias correction (BC)** | Yes | No | Yes | ST + TR |
| **BC Anchor** | SBUV/2, IASI[††], AIRS[‡‡] | N/A | SBUV/2, MLS | |
| **Quality control** | O-B < 30DU | N/A | O-B < 30DU | Mostly ST |
| **Forecast model & Chemistry** | Modified CD86 | Modified CD86 | | ST |
| | | | MOZART-3 CTM (Kinnison et al., 2007) | ST + TR |

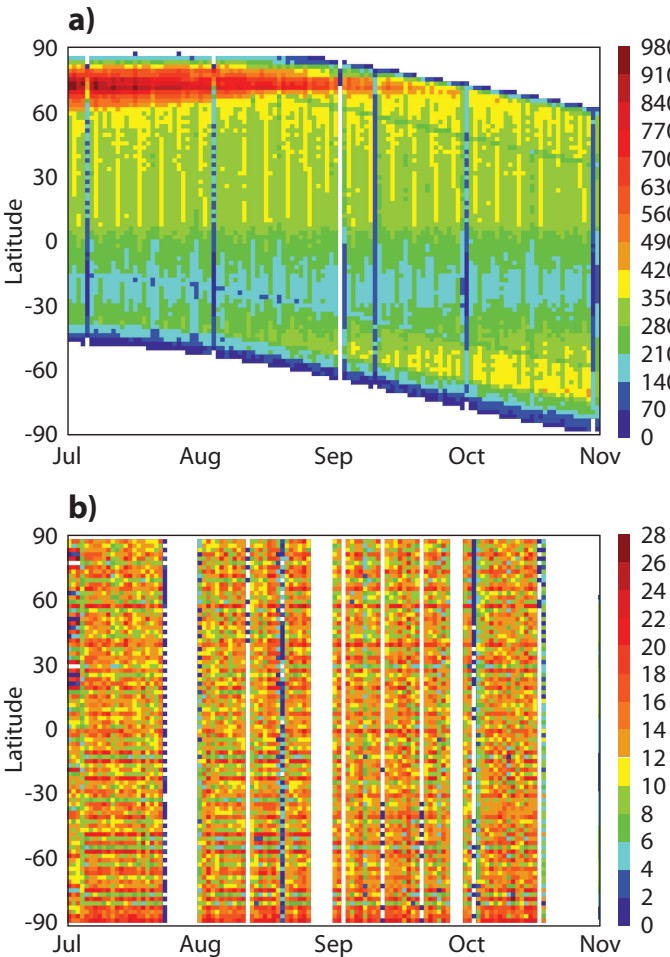

**Figure 1.** Data coverage and daily count binned in 2° latitudinal bands for the CCI GOME-2 (top panel) and the CCI MIPAS (bottom panel) data during the period July-October 2008. The two colour scales are different. The colour scale used for GOME-2 (panel **a**) varies from 0 (dark blue) to 980 (dark red) observations with a step of 70; the one used for MIPAS ranges from 0 (dark blue) to 28 (dark red) observations with a step of 2.

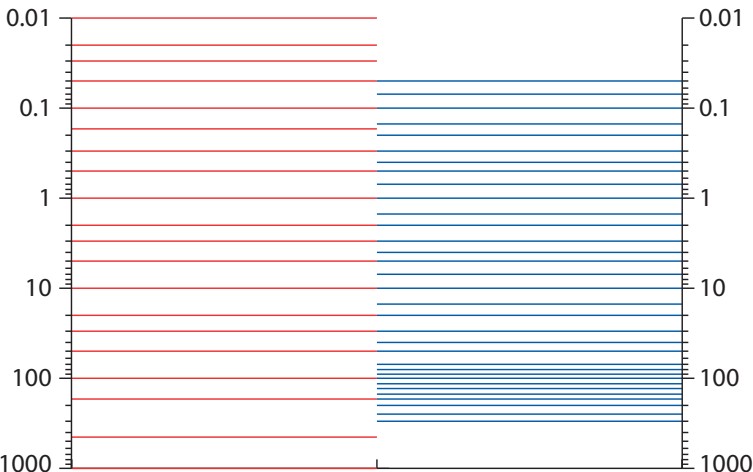

**Figure 2.** Schematic of the two ozone products' vertical coverage and vertical resolution as provided by the CCI GOME-2 (red lines) and the CCI MIPAS (blue lines) retrieval algorithms. The vertical axis represents pressure in hPa. The surface pressure level for GOME-2 is for illustration purposes drawn at 1000 hPa.

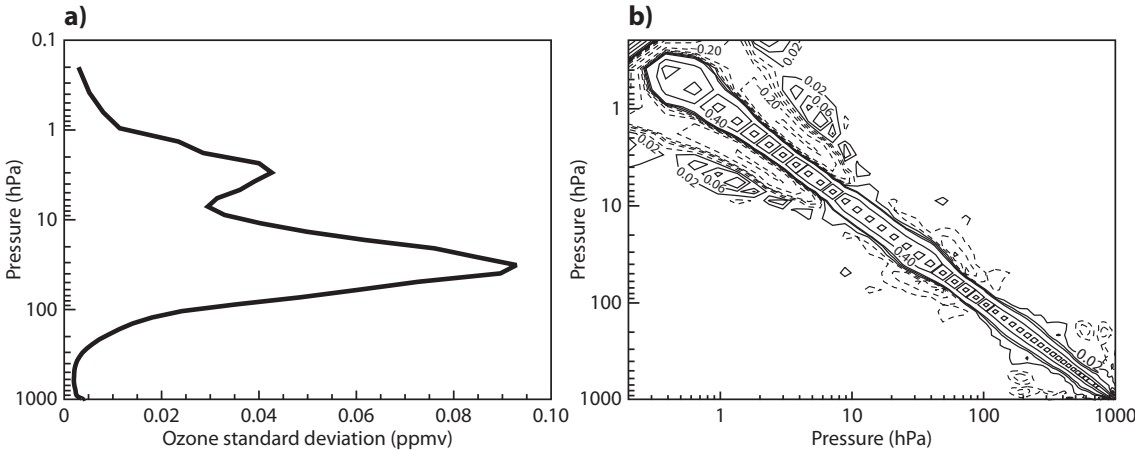

**Figure 3.** Example of ozone background error standard deviation profile (left panel) and vertical correlation matrix (right panel) for the ozone background errors. Data are in mixing ratio (parts per million by volume, ppmv). In the right panel, negative correlations are plotted as dashed lines, and the interval is 0.04 ppmv between -0.1 and 0.1 ppmv, and 0.2 for larger absolute values. Source Dragani and McNally (2013).

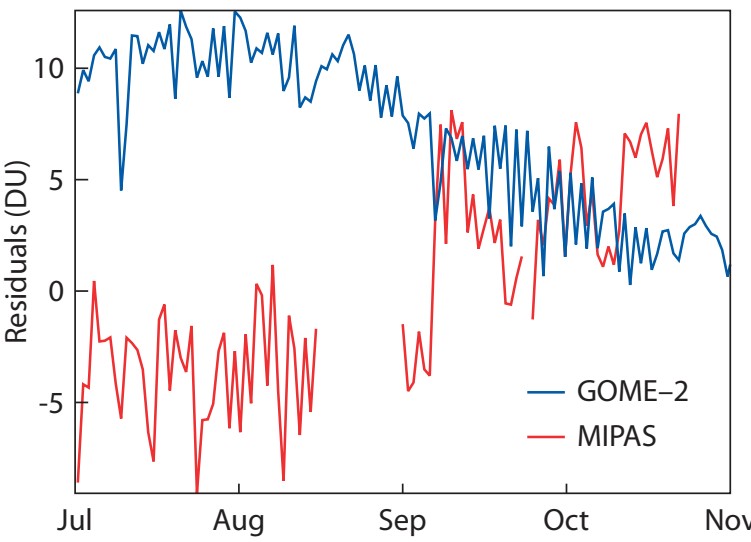

**Figure 4.** Time series of the global mean analysis departures for GOME-2 NPO$_3$ (blue line) and MIPAS LPO$_3$ (red line) over the period Jul-Oct 2008. Statistics are shown as integrated columns computed over the GOME-2 and MIPAS common vertical grid spanning from 0.05 to 100 hPa. Data are in DU.

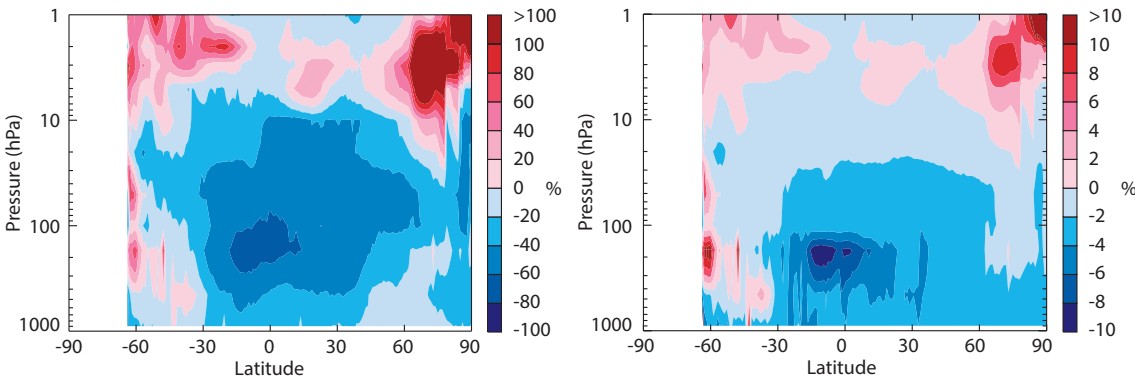

**Figure 5. Left panel:** Vertical cross-section of the difference between the estimated and the provided uncertainty relative to the provided uncertainty for GOME-2 NPO$_3$ over the period Jul-Oct 2008. Negative (positive) values in blue (red) colours mean that the provided uncertainty is larger (smaller) than that estimated with the Desroziers et al. (2005) method. Data are in %.**Right panel:** As in the left panel but with relative difference computed with respect to the observation instead of its provided uncertainty. Data are in %.

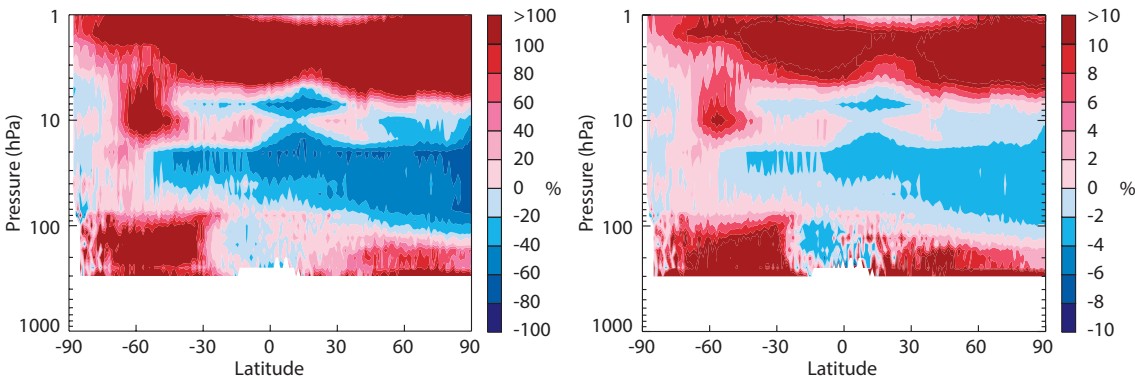

**Figure 6.** As in Figure 5 but for the CCI MIPAS LPO$_3$ data.

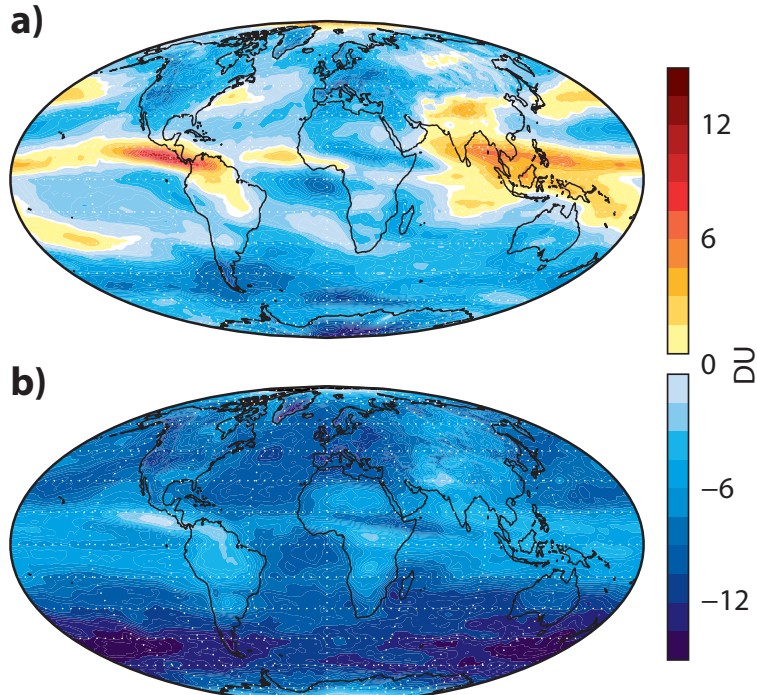

**Figure 7.** TCO$_3$ mean differences between the analyses from Exp/Ctrl and those from Exp/GOME2 (top) and Exp/MIPAS (bottom) computed for the period Aug-Oct 2008. Negative (positive) values in blue (red) colours mean that the additional instrument is increasing (decreasing) the TCO$_3$ amount compared to that obtained from the ozone datasets assimilated in Exp/Ctrl. Data are in DU. The colour scale ranges from -15 DU (dark blue) to +15 DU (dark red) with a step of 1.5 DU.

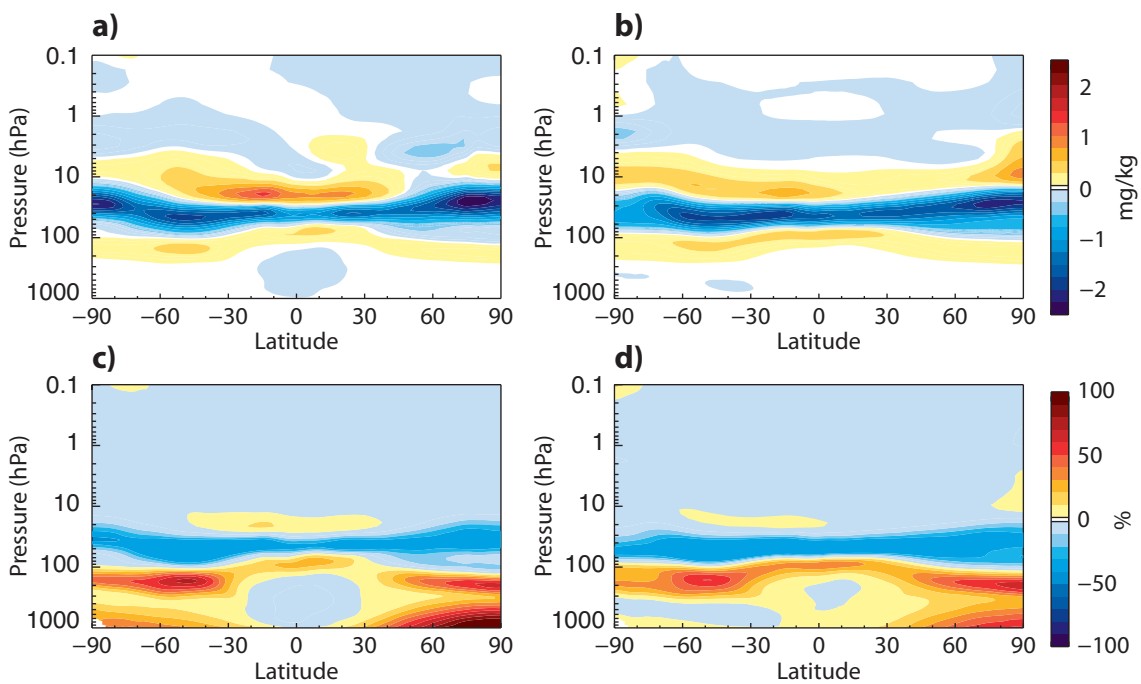

**Figure 8. Top panels:** Zonal mean temporal mean differences between the ozone analyses from Exp/Ctrl and the ozone analyses from Exp/GOME2 (top left panel) and Exp/MIPAS (top right panel) computed for the period Aug-Oct 2008. Negative (positive) values in blue (red) colours mean that the additional instrument is increasing (decreasing) the ozone mixing ratio amount compared to that obtained from the ozone datasets assimilated in Exp/Ctrl. The white regions are where the differences are within ± 0.1 mg/kg. Data are in mg/kg. The colour scale ranges from -2.5 mg/kg (dark blue colour) to +2.5 mg/kg (dark red colour) with a step of 0.25 mg/kg, with the exception of the yellow and pale blue that start at 0.1 and -0.1 mg/kg, respectively, and represent a step of 0.15 mg/kg. **Bottom panels:** Like in the top panels but for the relative differences computed with respect to a temporal mean global mean ozone profile obtained from the Exp/Ctrl analyses. Data are in %. The colour scale ranges from -100 % (dark blue colour) to +100 % (dark red colour) with a step of 10%.

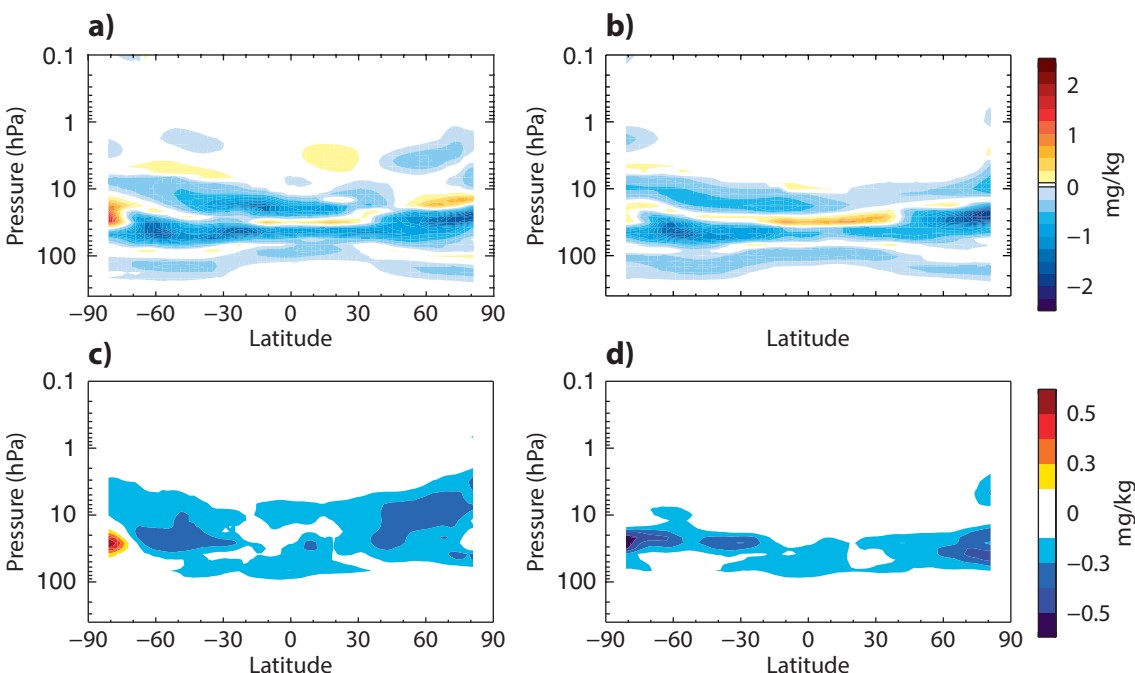

**Figure 9. Top panels:** Change in the zonal mean differences between the MLS retrievals and co-located ozone analyses from the perturbation experiment compared to the Exp/Ctrl for Aug-Oct 2008. The perturbation experiment is Exp/GOME2 in the left panel, and Exp/MIPAS in the right panel. Negative (positive) values in blue (red) colours indicate a decrease (an increase) in the mean in the Control and thus a degraded fit to MLS ozone profiles. Data are in mg/kg. The colour scale ranges from -2.5 mg/kg (dark blue colour) to +2.5 mg/kg (dark red colour) with a step of 0.25 mg/kg. **Bottom panels:** Like in the top panels but for the standard deviation of differences. The colour scale ranges from -0.6 mg/kg (dark blue colour) to +0.6 mg/kg (dark red colour) with a step of 0.1 mg/kg.

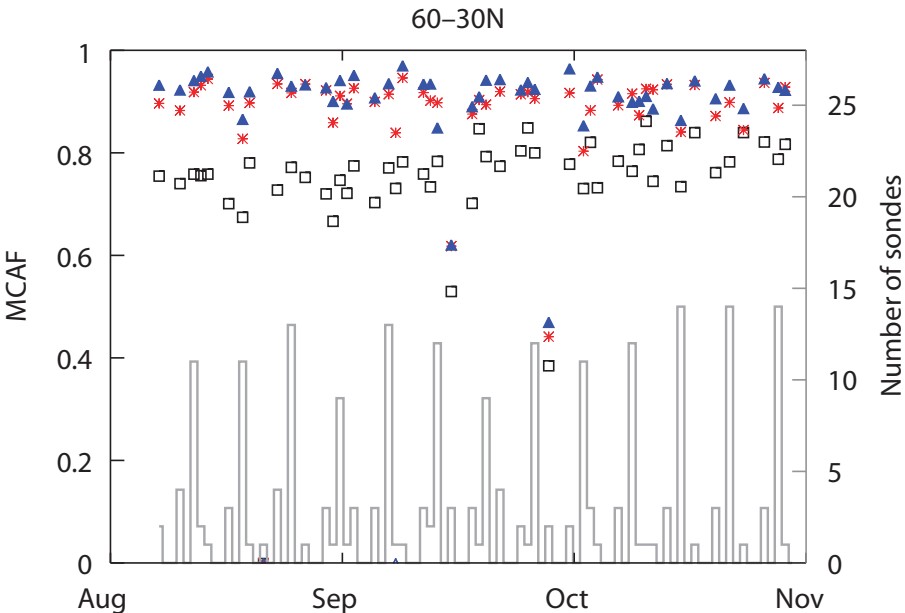

**Figure 10.** Time series of the stratospheric contribution (considered as that at pressure levels lower than 100 hPa) to the Mean Common Area Fraction (MCAF) score computed for different sonde profiles and the three sets of ozone analyses. The analyses were taken from Exp/Ctrl (black squares), and the two Exp/PERT experiments assimilating also the reprocessed GOME-2 $NPO_3$ (red asterisks) and MIPAS $LPO_3$ (blue triangles). The plot refers to the midlatitudes in the NH. The grey histogram in each panel shows the number of daily sondes included in the MCAF score and refers to the right hand side axis.

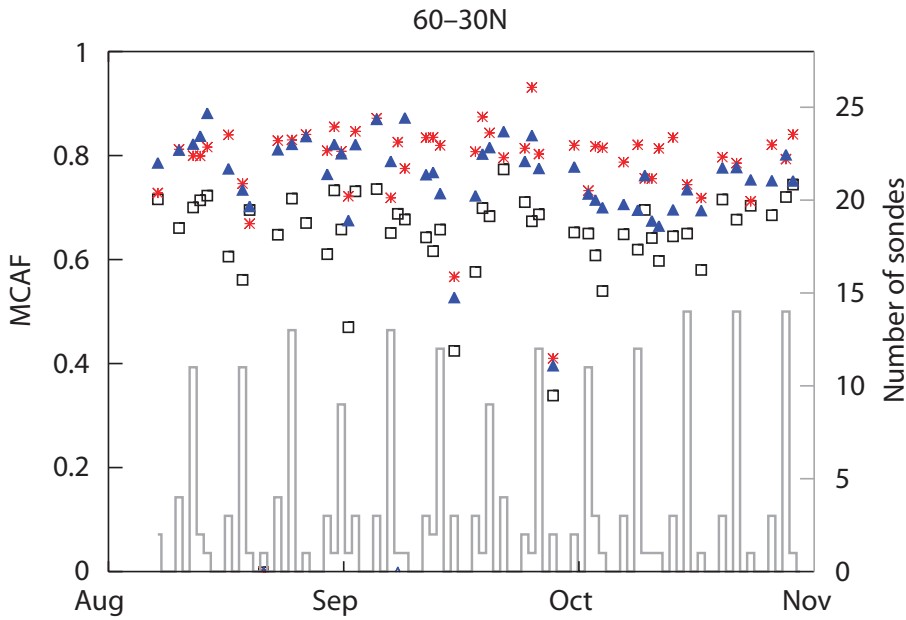

**Figure 11.** As Figure 10, but for the tropospheric contribution assumed as the contribution from the layer spanning the atmosphere from the surface up to 100 hPa.

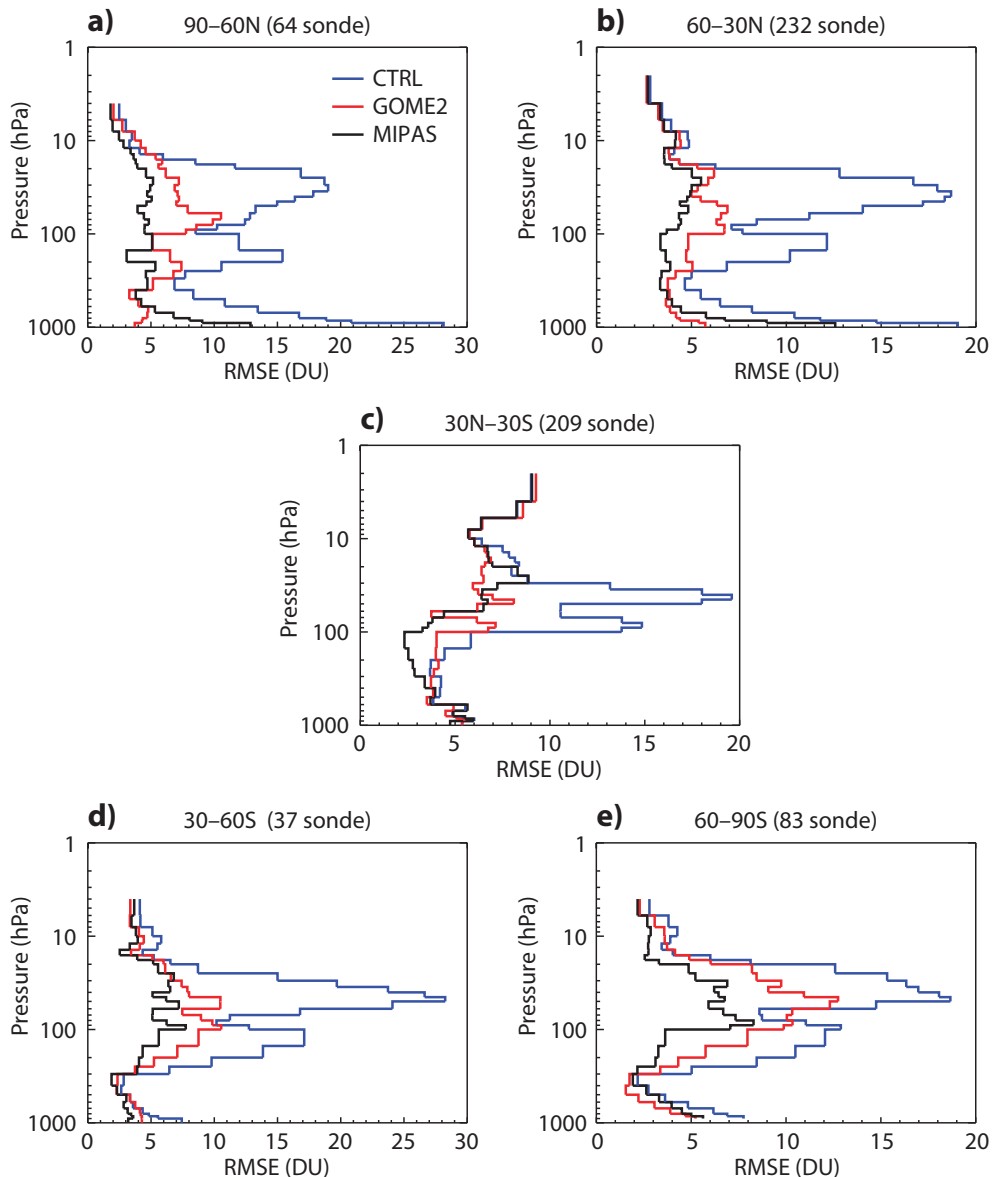

**Figure 12.** Fit of the ozone analyses from three experiments to ozone sondes given in terms of the RMSE over five latitudinal bands (one per panel as follows: 90°-60°N in panel **a**, 60°-30°N in panel **b**, 30°N-30°S in panel **c**, 30°-60°S in panel **d**, and 60°-90°S in panel **e**). The comparisons were computed by averaging over Aug - Oct 2008. The three analyses were taken from Exp/Ctrl (blue lines), and the two Exp/PERT experiments assimilating also the reprocessed GOME-2 $NPO_3$ (red lines) and MIPAS $LPO_3$ (black lines) from the O3-CCI. The latitudinal band each panel refers to and the number of ascents included in the average can be found in the corresponding panel title. Data are in Dobson Unit (DU).