# Peer review of "A comparative analysis of UV nadir-backscatter and infrared limb-emission ozone data assimilation"

_Atmospheric Chemistry and Physics, 2016_

## Referee Comment (RC3)

Review of the manuscript

**A comparative analysis of UV nadir-backscatter**
**and infrared limb-emission ozone data assimilation**

submitted to ACP by R. Dragani

S. Chabrillat, BIRA-IASB, May 2016

**General Comments**

I believe that this paper is a useful and valuable contribution to the field of ozone data assimilation but it fails to consider related work, and many appropriate references are missing. Hence while there is no need for any additional assimilation experiment, the text should still undergo major revisions.

1. This paper gives a false impression that ozone data assimilation is still in its infancy. There is a whole community working on this topic for a long time but none of its previous work is mentioned in the introduction nor considered in the discussion. I think that the introduction should be extended to provide proper context, and that this context should be used in the discussion.

2. The datasets assimilated in this study were developed for the ESA project O3-CCI. That project led to several validation papers which discuss extensively the uncertainties and information content of the corresponding datasets. Since a good evaluation of observational uncertainties is paramount for data assimilation, such prior work is highly relevant for this paper. Hence the O3-CCI validation papers should be at least cited in section 4, and the choices made for the present assimilation study should be discussed in this context.

3. The averaging kernels of GOME-2 nadir profiles are still not taken into account by IFS. This is a serious limitation of the present study, because many other assimilation systems now take properly into account such vertical smoothing errors. This limitation should at least be clearly stated in the conclusions and abstract of the paper: "*This study demonstrates the potentials and limitations of each dataset and instrument type*" – but only in the context of data assimilation with the current IFS at ECMWF.

**Specific Comments**

1. The introduction does not mention the results obtained in previous projects about data assimilation of stratospheric ozone, giving a false impression that ozone analyses are available only in two meteorological reanalyses (i.e. ERA-Interim and MERRA-2). Yet simultaneous assimilation of limb and nadir ozone datasets was reported and discussed as early as 2002 (Struthers et al., 2002). Nearly ten years ago, Lahoz et al. (2007) were already able to review this field. Even considering only the European projects, I believe that it is not possible to ignore such prior work as the ASSET intercomparison (Geer et al., 2006), the developments for the PROMOTE project (Viscardy et al., 2010) or the numerous results obtained for the MACC series of projects (see e.g. Inness et al., 2013; Inness et al., 2015). The absence of any citations about ozone assimilation in MACC is especially strange, because the MACC projects were coordinated

by the same Institution as the author (ECMWF) and relied on a version of the same model (IFS). MACC allowed an intercomparison of the ozone analyses delivered in Near Real-Time by four different systems assimilating nadir and/or limb datasets (Lefever et al., 2015). Even though the assimilation experiments were very different, this earlier study reached a very similar conclusion with a very similar Data Assimilation System (DAS): "*IFS-MOZART is able to deliver realistic analyses of ozone both in the troposphere and in the stratosphere, but this requires the assimilation of observations from nadir-looking instruments as well as the assimilation of profiles, which are well resolved vertically and extend into the lowermost stratosphere*". Overall it is necessary to extend significantly the introduction in order to provide the missing context, and to take prior work into consideration in the discussion of the results (section 5.2.2).

2. P.1, line 24: the concern for the ozone decline is primarily due to the expected increase of Ultraviolet radiation at the surface. This should be mentioned in the introduction, along with a general reference about the issue.

3. The description of the assimilated datasets (section 2) and the data quality analysis (section 4) both fail to consider the extensive validation work realized for the O3-CCI ozone datasets. At least three papers investigate the quality of the MIPAS and GOME-2 datasets which are assimilated here. Hassler et al. (2014) present an overview of stratospheric ozone profile measurement data, document measurement techniques, spatial and temporal coverage, vertical resolution, native units and measurement uncertainties; Laeng et al. (2015) and Keppens et al. (2015) compared the available retrieval algorithms for MIPAS and GOME-2, respectively, explaining the choice of the algorithms selected for the O3-CCI datasets. These studies about observational uncertainties should be used in the description of the assimilated datasets and could be useful for the discussion of the results. Miles et al. (2015) should be cited, not only as a reference for the assimilated GOME-2 dataset, but also for its specific validation results.

4. The limited vertical resolution of the GOME-2 dataset should be explained more extensively, citing a specific paper (e.g. Keppens et al., 2015) in addition to the overarching reference (Rodgers, 2000). Since GOME-2 profiles have "*between 5 and 6 degrees of freedom*", figure 2 does not show their vertical resolution. It shows instead the vertical grid of the retrieved product. This confusion could be seriously misleading for the novice reader. While it is less of a concern thanks to its limb-viewing geometry, MIPAS does not have perfect vertical resolution either (von Clarmann and Grabowski, 2006; Laeng et al., 2015). This should also be mentioned in section 2.

5. Description of the DAS (section 3): what is the IFS version number ("cycle") used here? How does it compare with the versions used in ERA-Interim (Dragani, 2011) and the MACC reanalysis (Inness et al., 2013) as far as ozone assimilation is concerned?

6. The modelling of ozone in IFS is not properly described, again leading to a lack of context for the discussion of the results. How is ozone photochemistry represented in the forecast model? Assuming that the parameterization by Cariolle and Teyssèdre (2007) is used here, this is not an explicit modelling of ozone photochemistry. So what does the sentence (p.5, line 15) "*In this forecast model and analysis system, ozone is fully integrated (Dethof and Hólm, 2004)*" mean

exactly? The parameterization by Cariolle and Teyssèdre has some limitations which should be stated as they could explain some of the assimilation results.

7. P. 5, lines 23-26: "*accounting for the vertical sensitivity of any retrieved product as provided by the data averaging kernels (AKs) is currently not possible in the IFS*". Please provide a reference for this limitation of IFS. Many other DAS now do take AKs into account, implementing a straightforward approach (explained e.g. by von Clarmann and Grabowski, 2006). Hence this limitation of IFS is a key caveat for this study because it limits the applicability of its findings to other DAS (see third major comment). "*With such an approximation, the vertical spread of the ozone information provided by the assimilated ozone observations depends on the background error variances and covariance (B) for ozone*". Please provide a reference about this. This approximation also fails to properly take vertical smoothing errors into account, and may constrain the analysis with *a priori* information contained in the retrievals. For example, in some viewing geometries the GOME-2 retrievals do not contain any usable information close to the surface.

8. Figures 5-6 could be quite interesting for the retrieval and validation communities who are not familiar with the estimation of observational errors allowed by data assimilation (i.e. the method by Desroziers et al., 2005). The attempt to explain this method (p.6, lines 28-32) is quite unclear, it should be re-written and expanded. P. 7, lines 18-19: "*the differences between provided and estimated uncertainties appear to be rather large*". This is an important result for the aforementioned communities (even though these uncertainties "*only represent up to about 4% of the observation values*"). Hence it should be shown, i.e. figures 5-6 should be expanded with similar latitude-pressure cross-sections showing the provided uncertainties and using the same color scale.

9. P. 7, lines 1-4: paragraph is unclear, please re-write. "*The reason for this is still under investigation at the time of writing*": indeed, this is not expected from the comparisons of ozone total columns between SBUV and GOME (Chiou et al., 2014)

10. P. 7, lines 9-11: "*The first-guess check implemented in the IFS discards all observations that, after successfully pass the data quality control, show discrepancies from the background of 30DU or more over the column*". Please re-write (e.g. "after having successfully passed"). "*Figure 4 shows that the observations from both instruments are well within such a threshold*". Figure 4 cannot be used to justify this Background Quality Check (BGQC) because it shows a global mean of the O-A departures while the BGQC is applied to individual observations.

**Minor Comments**

- P. 3, line 23: remove extra closing parenthesis.
- Legend of figure 1 : state the year plotted here (i.e. 2008)
- Figure 3: right panel would be much nicer as a color-coded contour plot.
- P. 7, line 6: "instruements" → "instruments"
- P. 7, line 9: "*that, after successfully pass the data quality control*" – please re-write

- P. 7, line 18: "differencies" → "differences"
- Table 1: Acronyms "BUV" and "ODS" are missing. I suggest to list first the satellite instruments, followed by the other acronyms
- Figure 7: color scale does not work well. Use same red-blue scale as for figures 5 and 6.

**Additional bibliographical references**

Chiou, E. W., Bhartia, P. K., McPeters, R. D., Loyola, D. G., Coldewey-Egbers, M., Fioletov, V. E., Van Roozendael, M., Spurr, R., Lerot, C., and Frith, S. M.: Comparison of profile total ozone from SBUV (v8.6) with GOME-type and ground-based total ozone for a 16-year period (1996 to 2011), Atmos. Meas. Tech., 7, 1681-1692, doi:10.5194/amt-7-1681-2014, 2014.

Cariolle, D. and Teyssèdre, H.: A revised linear ozone photochemistry parameterization for use in transport and general circulation models:multi-annualsimulations,Atmos.Chem.Phys.,7,2183–2196, doi:10.5194/acp-7-2183-2007, 2007.

Hassler, B., Petropavlovskikh, I., Staehelin, J., August, T., Bhartia, P. K., Clerbaux, C., Degenstein, D., Mazière, M. De, Dinelli, B. M., Dudhia, A., Dufour, G., Frith, S. M., Froidevaux, L., Godin-Beekmann, S., Granville, J., Harris, N. R. P., Hoppel, K., Hubert, D., Kasai, Y., Kurylo, M. J., Kyrölä, E., Lambert, J.-C., Levelt, P. F., McElroy, C. T., McPeters, R. D., Munro, R., Nakajima, H., Parrish, A., Raspollini, P., Remsberg, E. E., Rosenlof, K. H., Rozanov, A., Sano, T., Sasano, Y., Shiotani, M., Smit, H. G. J., Stiller, G., Tamminen, J., Tarasick, D. W., Urban, J., van der A, R. J., Veefkind, J. P., Vigouroux, C., von Clarmann, T., von Savigny, C., Walker, K. A., Weber, M., Wild, J., and Zawodny, J. M.: Past changes in the vertical distribution of ozone – Part 1: Measurement techniques, uncertainties and availability, Atmos. Meas. Tech., 7, 1395-1427, doi:10.5194/amt-7-1395-2014, 2014.

Hubert, D., Lambert, J.-C., Verhoelst, T., Granville, J., Keppens, A., Baray, J.-L., Cortesi, U., Degenstein, D. A., Froidevaux, L., Godin-Beekmann, S., Hoppel, K. W., Kyrölä, E., Leblanc, T., Lichtenberg, G., McElroy, C. T., Murtagh, D., Nakane, H., Russell III, J. M., Salvador, J., Smit, H. G. J., Stebel, K., Steinbrecht, W., Strawbridge, K. B., Stübi, R., Swart, D. P. J., Taha, G., Thompson, A. M., Urban, J., van Gijsel, J. A. E., von der Gathen, P., Walker, K. A., Wolfram, E., and Zawodny, J. M.: Ground-based assessment of the bias and long-term stability of fourteen limb and occultation ozone profile data records, Atmos. Meas. Tech. Discuss., 8, 6661-6757, doi:10.5194/amtd-8-6661-2015, 2015.

Inness, A., Baier, F., Benedetti, A., Bouarar, I., Chabrillat, S., Clark, H., Clerbaux, C., Coheur, P., Engelen, R. J., Errera, Q., Flemming, J., George, M., Granier, C., Hadji-Lazaro, J., Huijnen, V., Hurtmans, D., Jones, L., Kaiser, J. W., Kapsomenakis, J., Lefever, K., Leitão, J., Razinger, M., Richter, A., Schultz, M. G., Simmons, A. J., Suttie, M., Stein, O., Thépaut, J.-N., Thouret, V., Vrekoussis, M., Zerefos, C., and the MACC team: The MACC reanalysis: an 8 yr data set of atmospheric composition, Atmos. Chem. Phys., 13, 4073–4109, doi:10.5194/acp-13-4073-2013, 2013.

Inness, A., Blechschmidt, A.-M., Bouarar, I., Chabrillat, S., Crepulja, M., Engelen, R. J., Eskes, H., Flemming, J., Gaudel, A., Hendrick, F., Huijnen, V., Jones, L., Kapsomenakis, J., Katragkou, E.,

Keppens, A., Langerock, B., de Mazière, M., Melas, D., Parrington, M., Peuch, V. H., Razinger, M., Richter, A., Schultz, M. G., Suttie, M., Thouret, V., Vrekoussis, M., Wagner, A., and Zerefos, C.: Data assimilation of satellite-retrieved ozone, carbon monoxide and nitrogen dioxide with ECMWF's Composition-IFS, Atmos. Chem. Phys., 15, 5275-5303, doi:10.5194/acp-15-5275-2015, 2015.

Keppens, A., Lambert, J.-C., Granville, J., Miles, G., Siddans, R., van Peet, J. C. A., van der A, R. J., Hubert, D., Verhoelst, T., Delcloo, A., Godin-Beekmann, S., Kivi, R., Stübi, R., and Zehner, C.: Round-robin evaluation of nadir ozone profile retrievals: methodology and application to MetOp-A GOME-2, Atmos. Meas. Tech., 8, 2093-2120, doi:10.5194/amt-8-2093-2015, 2015.

Laeng, A., D. Hubert, T. Verhoelst, T. von Clarmann, B.M. Dinelli, A. Dudhia, P. Raspollini, G. Stiller, U. Grabowski, A. Keppens, M. Kiefer, V. Sofieva, L. Froidevaux, K.A. Walker, J.-C. Lambert, C. Zehner, The ozone climate change initiative: Comparison of four Level-2 processors for the Michelson Interferometer for Passive Atmospheric Sounding (MIPAS), Remote Sensing of Environment, Volume 162, 316-343, doi: 10.1016/j.rse.2014.12.013, 2015.

Lefever, K., van der A, R., Baier, F., Christophe, Y., Errera, Q., Eskes, H., Flemming, J., Inness, A., Jones, L., Lambert, J.-C., Langerock, B., Schultz, M. G., Stein, O., Wagner, A., and Chabrillat, S.: Copernicus stratospheric ozone service, 2009–2012: validation, system intercomparison and roles of input data sets, Atmos. Chem. Phys., 15, 2269-2293, doi:10.5194/acp-15-2269-2015, 2015.

Struthers, H., Brugge, R., Lahoz, W. A., O'Neill, A. and Swinbank, R.: Assimilation of ozone profiles and total column measurements into a global general circulation model, J. Geophys. Res., 107(D20), 4438, doi:10.1029/2001JD000957, 2002.

Viscardy, S., Q. Errera, Y. Christophe, S. Chabrillat and J. C. Lambert: Evaluation of Ozone Analyses From UARS MLS Assimilation by BASCOE Between 1992 and 1997, IEEE Journal of Selected Topics in Applied Earth Observations and Remote Sensing, vol. 3, no. 2, pp. 190-202, doi:10.1109/JSTARS.2010.2040463, 2010.

von Clarmann, T. and Grabowski, U.: Elimination of hidden a priori information from remotely sensed profile data, Atmos. Chem. Phys., 7, 397-408, doi:10.5194/acp-7-397-2007, 2007.

---

## Referee Comment (RC1) · Anonymous Referee #1 · 24 Mar 2016

**Review of the manuscript 'A comparative analysis of UV nadir-backscatter and infrared limb-emission ozone data assimilation' by Rossana Dragani**

The study analyses the results of including GOME-2 and MIPAS ozone retrievals in the ECMWF data assimilation system on top of the ozone data routinely assimilated in the IFS. The two perturbation experiments (with GOME-2 and MIPAS data included separately) are compared with the baseline IFS ozone and independent data from Aura MLS and ozonesondes. In both cases the inclusion of new profile data leads to improvements, particularly in the lower stratosphere and in the troposphere. The latter is true even in the case of MIPAS despite its limited vertical coverage. It is found that the differences between the perturbed and control experiments are more localized in the vertical for the MIPAS experiment, as expected given sharper averaging kernels associated with limb measurements, compared to the nadir UV-Vis methodology.

The subject matter is important for the chemical data assimilation community and fits the ACP profile perfectly. It is relevant to both, future reanalyses and near real time analyses going into the future (nadir UV). I find the paper to be well written with clear structure and compelling results. I especially like the MLS and ozonesonde comparisons results summarized in Figs. 9-12. The added value of MIPAS and GOME-2 data are clearly shown there.

I recommend the paper for publication in ACP subject to very few very minor revisions as delineated below.

**General comment**

Reading back through Dragani 2011, MIPAS ozone (different version) and GOME (ERS-2, not the same GOME) were already assimilated in ERA Interim. I know it's not the same data but I think it should be mentioned somewhere. How do the results presented here compare with ERA Interim? Does the CCI MIPAS ozone bring anything new compared to the version that was used in ERA-I? I'm not asking for any extensive comparisons, just a comment.

**Specific comments**

P2 L24. McCarty et al. is still not finished. At this point I suggest changing this reference to Bosilovich et al. (2015). The ozone chapter contains the same information and this tech memo is already published and citable.

P4 L 5-14. What is GOME's footprint?

P4 L26. The link is old and redirects to http://cci.esa.int/. You may want to update it

P6 L7 'the ozone continuity equation is expressed as a linear relaxation…' Hmm, there's more to the continuity equation than just chemistry. How about 'contains' instead of 'is expressed as'? Maybe I misunderstood something.

P9 Last paragraph of Section 5.1. Is this because with the stratosphere constrained by MIPAS the analysis increments arising from total column data are distributed differently? You talk about this later on (P13) – how about 'this will be discussed in Section 5.2.2'? Also, see my comment to P13 L6.

P10 L1. 'version 3.04', is this correct? As far as I know the recent 'official' versions are 2.2, 3.3, 3.4 and 4.2

P10 L19. A bit more about how the 'degrading' is done. Is it by interpolation from the two nearest pressures or the average within the layer onto which you are interpolating? This probably makes little difference for MLS comparisons but I found that for ozonesonde data, with their high vertical resolution (many sonde measurements per model layer) it's better to integrate than to interpolate between the two nearest points. This is because the model/DAS layer ozone values represent the layer averages, whereas sondes provide point measurements.

P11 L20-29. It would help to see some percent values. Not necessarily in the figure but in the text.

P13 L6. I agree that this is the most probable reason why the MIPAS analysis results are improved below 400 hPa but it is not really shown here – it is just stated. The reader may wander if vertical transport between the observed and unobserved layers wouldn't also play a role.

P15 L3. Again, I'm confused about the MLS data version (3.04 or 3.3?)

**Technical comments**

P10 L3. '82S to 82N' → '82°S to 82°N'

P11 L32 'southern than' → 'south of'

P12 L25 'what does it happen in the lowermost troposphere' → 'what happens'?

**Reference**

Bosilovich M. G. and co-authors 2015: MERRA-2: Initial evaluation of the climate. Technical Report Series on Global Modeling and Data Assimilation, NASA/TM–2015-104606/Vol. 43. Available from http://gmao.gsfc.nasa.gov/reanalysis/MERRA-2/docs/

---

## Referee Comment (RC2) · Anonymous Referee #2 · 5 Apr 2016

**Review of the paper titled "A comparative analysis of IV nadirbackscatter and infrared limb-emission ozone data assimilation" by Dragani.**

In this paper, the author investigates the impact of the addition of a nadir and a limb ozone profile data set, produced by ESA's O3-CCI, on the ozone fields in the ECMWF IFS Data Assimilation System. The global distribution of ozone and the change in uncertainty are studied and compared to a reference without additional ozone information. The two new data sets are also compared to a control run, and to external reference data (e.g.: MLS and ozone sonde). The author demonstrates that both the nadir and the limb ozone profile data sets can improve the assimilation, and makes a case for more limb measurements when most of the satellite instruments to be launched in the near future are of the nadir class.

**General Comments**

In this paper I see the name of the satellite instrument and the retrieved ozone product(s) being used interchangeably a lot when they should be kept separate in my opinion. For example: P4, L1-3: In these sentences the two are starting to get intermixed.

While the instrument is unique, there are several implementations of retrieval algorithms for a particular satellite instrument's data. For example, for GOME-2 the ozone retrieval algorithms / products known are Miles et al (O3-CCI), Cai et al, and Hassinen et al (O3MSAF). The retrieval algorithms may all have different behaviour, which makes it harder to make general statements like 'GOME-2 data is....'. The same holds for MIPAS, the instrument (and L1 data) is not the same as the ozone product coming out of a retrieval algorithm. Please check the manuscript, and identify where you really mean the instrument, and where you refer to the ozone retrieval product.

**Specific Comments**

P1, L23: You mention the warming/cooling of the air in the atmosphere, and then mention its (long term) effect on climate. A warming / cooling of air has a more immediate effect on the atmosphere: a temperature difference leads to a density difference, which leads to a pressure difference, which in turn leads to flow of air. In that way, the global ozone distribution can affect the dynamics of the atmosphere.

P4, L12: Do you mean resolution, or sampling? The sampling of the instrument is usually defined as the distance between detector pixels (in nm), but the resolution of the instrument is also affected by the width of the instrument's slit function (which may be wider, and span more than one detector pixels).

P7, L22-25: The author starts off with mentioning that O3-CCI/GOME-2 has the largest difference in the 4 month period. While this is true in the beginning, it would be more insightful for the reader if the discussion on the differences would be split into the first two months and the last two months, as is the case in the later sentences where the O3-CCI/MIPAS differences are split into Jul/Aug

and Sept/Oct. Given that the behaviour of the difference changes with time I feel that giving a range of the average difference is more representative than a single value over the four month period.

P8, L8-11: Using the larger provided uncertainty to compensate for the fact that vertical correlations (by means of Averaging Kernels (AK's) and covariance matrices from the ozone retrieval) are not used in the assimilation systems is risky. Can the author give an estimate whether the larger provided retrieval uncertainty is similar in the value and the sign of the vertical correlations?

P8, L18: The author states that no corrections are applied for GOME-2 O3 nadir profile data above 5hPa but does not mention what kind of corrections are applied below. It leaves the reader in doubt on what happens. Only at P8-L32-34, at the end of the paragraph that discusses MIPAS, the reader finally finds that no corrections are applied to either the nadir or limb retrieval product. Please make this clear in the paragraph that discusses O3-CCI/GOME-2.

P9, L5-6: The author mentions that the behaviour in the tropical region is different for O3-CCI/GOME-2 than for O3-CCI/MIPAS. From figure 4 it is clear that the largest differences seem to coincide with the ITCZ, which is a clouded region in the tropics with high cloud tops. In P5-L9+10 the author states that the MIPAS data has been carefully screened for clouds, while I see no such statement for the O3-CCI/GOME-2 data. This could explain the large differences, when the nadir ozone profile retrievals are affected by ozone ghost columns, as the clouds block the observation of ozone below the cloud top. Would it be possible to investigate the effect of removal of pixels with a large cloud fraction from the O3-CCI/G2 data set on the global distribution of differences?

P10, L19: As far as I understand the comparison of two satellite retrievals, both instruments should be brought to a common grid and the AK's should be cross-applied. See Calisesi, et al (2005), Regridding of remote soundings: Formulation and application to ozone profile comparison, J. Geophys. Res., doi:10.1029/2005JD006122. Would this be a feasible approach for this study? For comparisons using reference data with a very high vertical resolution, such as ozone sondes, the transformation in Calisesi is not required because the ozone sonde's 'averaging kernel' peaks only near the measurement altitude (as it is a very localised measurement of the air it passes through).

P11, L29-31: Both instruments show reductions of the standard deviation (Fig 9). The one from O3-CCI/GOME-2 occurs over a wider vertical range than the one from O3-CCI/MIPAS, wheras the latter seems to have stronger localised reductions. Which of the two would be preferable for the assimilation as a whole and why?

**General question**

The author demonstrates that the comparison with MLS and ozone sondes improves when GOME-2 and MIPAS based ozone profiles are assimilated, but if it is not too far out of scope of this paper, it would be interesting to get an indication of the change in the skill of the IFS in general as a result of the assimilation of the additional ozone input. E.g.: the effect on wind vectors or temperature.

**Typographical comments**

P2, L2: signature  $\rightarrow$  ... the **signing** of an international treaty ... [signature is a noun, signing is the activity].

P3, L5: greatly  $\rightarrow$  very

P4, L28: The CCI ...

P5, L10: verified? You may mean 'present'.

P5, L32: satellite**s** (plural).

P6, L16: "An example of **a** background error profile and **a** vertical correlation..."

P6, L18: Introduction of acronym TCO3 without prior explanation (also not in table 1).

P12, L24: is equivalent

P12, L25: what  $\rightarrow$  why

Figure 12: The plots are small and the black and blue are sometimes hard to distinguish with this line width. Would it be possible to provide larger plots with thicker line? One could try a 2-1-2 panel ordering instead of the current 3-2.

References: Miles et al (2015): double doi Munro et al (2006): Please check initials of Munro, it seems that there are spurious letters, as the other reference has an 'R.' only.

---

## Author Comment (AC1) · 20 Jun 2016

Please see the supplement zip file for the final author comments.

Please also note the supplement to this comment:
http://www.atmos-chem-phys-discuss.net/acp-2016-96/acp-2016-96-AC1-supplement.zip

---

## Author Response (AR1)

**Reply to comments to the paper "A comparative analysis of UV nadir-backscatter and infrared limb-emission ozone data assimilation" (R. Dragani)**

I thank the Reviewers for their comments and suggestions. These have reported below (highlighted) with my reply (in normal text) and addressed in the current version of the manuscript where appropriate, unless indicated otherwise.

A modified version of the ACPD paper with tracked changes is attached at the end of this document.

**Reply to Reviewer # 1:**

**General comments:**

Reading back through Dragani 2011, MIPAS ozone (different version) and GOME (ERS-2, not the same GOME) were already assimilated in ERA Interim. I know it's not the same data but I think it should be mentioned somewhere. How do the results presented here compare with ERA Interim? Does the CCI MIPAS ozone bring anything new compared to the version that was used in ERA-I? I'm not asking for any extensive comparisons, just a comment.

The Reviewer is right in saying that ERS-2 GOME ozone profiles and ENVISAT MIPAS ozone profiles were both assimilated in the ERA-Interim production. In particular, it is noted that:

- The CCI algorithm used to retrieve the GOME-2 ozone profiles used in the present study is a development of the Rutherford Appleton Laboratory (RAL) retrieval scheme that was used to retrieve the ERS-2 GOME ozone profiles assimilated in the ERA-Interim reanalysis. This point has now been added to the paper.
- Regarding the MIPAS data, there are three differences (in the spectral characteristics of the Level 1b data, in the data processing, in the L2 algorithm) between the CCI product and the one used in ERA-Interim. These have now been discussed in the paper.

**Specific comments:**

- P2 L24: McCarty et al. is still not finished. At this point I suggest changing this reference to Bosilovich et al. (2015). The ozone chapter contains the same information and this tech memo is already published and citable.
  Reference replaced as advised.

- P4 L 5-14: What is GOME's footprint?
  For the period used in this study (2008), the GOME-2 typical footprint in the forward scan is 80 km × 40 km (across track × along track). This aspect has now been added to the paper. It is noted here that after the launch of MetOp-B, a change in the orbit swath (from 1920 km to 960 km) was applied to MetOp-A (July 2013) that led to a smaller footprint of 40 km × 40 km.

- P4 L26: The link is old and redirects to http://cci.esa.int/. You may want to update it.
  The link has been updated, as advised.

- P6 L7: 'the ozone continuity equation is expressed as a linear relaxation...' Hmm, there's more to the continuity equation than just chemistry. How about 'contains' instead of 'is expressed as'? Maybe I misunderstood something.
  The sentence has now been rephrased.

- P9 Last paragraph of Section 5.1: Is this because with the stratosphere constrained by MIPAS the analysis increments arising from total column data are distributed differently? You talk about this later on (P13) – how about 'this will be discussed in Section 5.2.2'? Also, see my comment to P13 L6.
  A reference to section 5.2.2 has been added, as advised.

- P10 L1: 'version 3.04', is this correct? As far as I know the recent 'official' versions are 2.2, 3.3, 3.4 and 4.2.
  The Reviewer is right, the reported MLS version was incorrect. From the original hdf files, the MLS data used in the study is version 3.3. This has now been corrected throughout.

- P10 L19: A bit more about how the 'degrading' is done. Is it by interpolation from the two nearest pressures or the average within the layer onto which you are interpolating? This probably makes little difference for MLS comparisons but I found that for ozonesonde data, with their high vertical resolution (many sonde measurements per model layer) it's better to integrate than to interpolate between the two nearest points. This is because the model/DAS layer ozone values represent the layer averages, whereas sondes provide point measurements.
  The model ozone and the sonde ozone profiles are on two different pressure level grids at the start of the procedure. As it was mentioned in the paper, the comparison is performed using a vertical interpolation on the coarsest vertical grid, which for the ozone sondes is represented by the model levels. Please note that this interpolation is only performed within the region of the atmosphere covered by both profiles, i.e. no extrapolation is performed as part of the procedure, and only applied to the sondes that reach at least 40 hPa. Once the model and sonde profiles are on the same vertical grid, integrated columns are computed for both. This vertical integration is only applied for plotting purposes to facilitate the visual inspection. This point has now been clarified in the paper.

- P11 L20-29: It would help to see some percent values. Not necessarily in the figure but in the text.
  Percent values have been added to the text, as advised.

- P13 L6: I agree that this is the most probable reason why the MIPAS analysis results are improved below 400 hPa but it is not really shown here – it is just stated. The reader may wander if vertical transport between the observed and unobserved layers wouldn't also play a role.
  I would exclude that the vertical transport could be in part responsible for the positive impact the assimilation of MIPAS has in the lower troposphere. Figure 8 panel d) in the paper presents the relative differences between the Exp/MIPAS and Exp/Ctrl analyses. It clearly shows changes in the ozone analyses below 400hPa down to the surface. This difference can only be triggered by the element(s) that differ in the two experiment setup, and that is the assimilation of MIPAS data, which has a vertical coverage down to 400 hPa at best. If the ozone assimilation was based on a multi-variate system (i.e. completely interactive) within a long-window 4D-Var, then perhaps a

change in the stratospheric ozone could have modified the circulation and produce an impact in the lower troposphere. But here, ozone is univariate. It means it has no impact on the rest of the system in general, and on the winds in particular (neither horizontal winds nor the vertical component). The vertical velocity in the upper troposphere down to 400hPa has values of up to 2-3 hPa/hour, which means the vertical transport could explain less than 50hPa vertical displacement within the 12 hour assimilation window, but not changes below 500hPa. A comment has been added to the paper.

- P15 L3: Again, I'm confused about the MLS data version (3.04 or 3.3?).
  This has now been corrected to version 3.3, thank you. See also reply above.

**Technical comments:**

- P10 L3: '82S to 82N' ➜ '82°S to 82°N'.
  Corrected as advised, thank you.
- P11 L32: 'southern than' ➜ 'south of'.
  Corrected as advised, thank you.
- P12 L25: 'what does it happen in the lowermost troposphere' ➜ 'what happens'?
  Corrected as advised, thank you.

**Reply to Reviewer # 2:**

**General comments:**

In this paper I see the name of the satellite instrument and the retrieved ozone product(s) being used interchangeably a lot when they should be kept separate in my opinion. For example: P4, L1-3: In these sentences the two are starting to get intermixed.

I do agree with the Reviewer that for simplicity the name of the instrument is often used to identify the dataset. In the example mentioned (namely P4, L1-3), I agree the two should not have been used interchangeably, and it has now been corrected. The paper has been reviewed for similar cases.

While the instrument is unique, there are several implementations of retrieval algorithms for a particular satellite instrument's data. For example, for GOME-2 the ozone retrieval algorithms / products known are Miles et al (O3-CCI), Cai et al, and Hassinen et al (O3MSAF). The retrieval algorithms may all have different behaviour, which makes it harder to make general statements like 'GOME-2 data is....'. The same holds for MIPAS, the instrument (and L1 data) is not the same as the ozone product coming out of a retrieval algorithm. Please check the manuscript, and identify where you really mean the instrument, and where you refer to the ozone retrieval product.

The Reviewer is correct in saying that for both the GOME-2 and the MIPAS instruments several algorithms have been developed. In the paper, having used the CCI datasets for both instruments, it felt safe to eventually omit the reference to CCI and simply identify each dataset by the corresponding instrument name. To avoid confusion, a note has now been added in section 2.

**Specific comments:**

- P1 L23: You mention the warming/cooling of the air in the atmosphere, and then mention its (long term) effect on climate. A warming / cooling of air has a more immediate effect on the atmosphere: a temperature difference leads to a density difference, which leads to a pressure difference, which in turn leads to flow of air. In that way, the global ozone distribution can affect the dynamics of the atmosphere.
  The point I was trying to make is that ozone can have an impact on different time scales, affecting the dynamics of the atmosphere as point out by the Reviewer, but also on longer time scale behaving as a green-house gas in the troposphere, thus impacting the Earth's climate. I rephrased it to try and make the point clearer.

- P4 L 12: Do you mean resolution, or sampling? The sampling of the instrument is usually defined as the distance between detector pixels (in nm), but the resolution of the instrument is also affected by the width of the instrument's slit function (which may be wider, and span more than one detector pixels).
  I indeed meant spectral resolution being between 0.2-0.4nm, e.g. from EUMETSAT note at www.eumetsat.int/website/home/Satellites/CurrentSatellites/Metop/MetopDesign/GOME2/index.html.

- P7 L22-25: The author starts off with mentioning that O3-CCI/GOME-2 has the largest difference in the 4 month period. While this is true in the beginning, it would be more insightful for the reader if the discussion on the differences would be split into the first two months and the last two

months, as is the case in the later sentences where the O3-CCI/MIPAS differences are split into Jul/Aug and Sept/Oct. Given that the behaviour of the difference changes with time I feel that giving a range of the average difference is more representative than a single value over the four month period.

In that sentence, I was pointing out that on average over the four month period, the analysis departures from the GOME-2 data showed a positive mean value around 5DU while those for MIPAS are on average close to zero. The discussion of figure 4 is done already by looking at the two periods separately as suggested by the Reviewer, from the paper: "*The MIPAS measurements indicate that during July-August the global mean ozone analyses are about 5DU too high (they were 10DU too low based on the GOME-2 data). Here, the discrepancy between the two instruments is very likely related to different coverage of the two instruments, particularly over the high latitudes in the SH, as shown in Figure 1. During September-October, the O-A residuals for the two instruments are more similar and they both indicate an underestimation of the ozone analyses of about 5DU above 100hPa*". If I correctly understood the Reviewer comment, I believe the discussion is already in line with their suggestion, so no change has been made to the text.

- P8 L8-11: Using the larger provided uncertainty to compensate for the fact that vertical correlations (by means of Averaging Kernels (AK's) and covariance matrices from the ozone retrieval) are not used in the assimilation systems is risky. Can the author give an estimate whether the larger provided retrieval uncertainty is similar in the value and the sign of the vertical correlations?

Regarding the use of the AKs (see also my reply to Reviewer # 3), in IFS the use of the AKs has not yet been implemented, thus using them is not an option. In the MACC/CAMS IFS (please note this is a different system from the operational IFS) some tests have been performed. However, the results obtained when assimilating the ozone data with the AKs compared to the same assimilation without AKs are at best neutral (A. Inness, personal communication). The use of the full error covariance matrix has never been tested, neither in IFS nor in the MACC/CAMS IFS. I would like to stress that the observation uncertainties used in the assimilation runs were those provided by the data providers, and that they were not inflated. What the paper shows is that the provided uncertainty seems to be larger than that diagnosed with the Desrozier method, which is based on first-guess and analysis departures. Also, please note that assimilating data with an inflated observation uncertainty is not risky. On the contrary, inflation is a standard practice used in data assimilation for safety and to be conservative, as with a larger uncertainty one can limit the level of changes the observations produce on the resulting analyses.

- P8 L18: The author states that no corrections are applied for GOME-2 O3 nadir profile data above 5hPa but does not mention what kind of corrections are applied below. It leaves the reader in doubt on what happens. Only at P8-L32-34, at the end of the paragraph that discusses MIPAS, the reader finally finds that no corrections are applied to either the nadir or limb retrieval product. Please make this clear in the paragraph that discusses O3-CCI/GOME-2.

As stated in the paper, data assimilation is normally performed in a conservative way, meaning observations are never given more weights than their uncertainties would imply. In the case of GOME-2, the comparison between the provided and estimated uncertainty shows that the former is larger than the latter below 5hPa, implying that the data assimilation would give a smaller weight (importance) to the observations than we would have anticipated from the uncertainty

estimate, thus here there is no need to further e.g. inflate the observation uncertainty. As stated in the paper, "…*The only consequence of such an overestimation would be to limit the impact of the corresponding data*". An explicit note has now been added to the paper.

- P9 L5-6: The author mentions that the behaviour in the tropical region is different for O3-CCI/GOME-2 than for O3-CCI/MIPAS. From figure 4 it is clear that the largest differences seem to coincide with the ITCZ, which is a clouded region in the tropics with high cloud tops. In P5-L9+10 the author states that the MIPAS data has been carefully screened for clouds, while I see no such statement for the O3-CCI/GOME-2 data. This could explain the large differences, when the nadir ozone profile retrievals are affected by ozone ghost columns, as the clouds block the observation of ozone below the cloud top. Would it be possible to investigate the effect of removal of pixels with a large cloud fraction from the O3-CCI/G2 data set on the global distribution of differences? I found in the literature a mention to the fact that in the retrieval scheme, cloud radiative transfer is not modelled explicitly. Instead an effective Lambertian surface albedo is co-retrieved. This is thought to have consequences below the cloud top (Miles et al, 2015). This information and the possible issues associated to not having an explicit cloud model have been included in the paper. The pixel cloud fraction is not available so it is not possible to check and remove cloud-affected pixels. I am not an expert in retrieval and it is difficult for me to say what impact modelling the clouds could have on the vertical profile, but I would think that the impact can mostly affect the lower troposphere up to the cloud top, and only marginally, if at all, the stratosphere. Figure 4 shows the global analysis departures over the layer 0.05-100 hPa (essentially the stratosphere and lower mesosphere).

- P10 L19: As far as I understand the comparison of two satellite retrievals, both instruments should be brought to a common grid and the AK's should be cross-applied. See Calisesi, et al (2005), Regridding of remote soundings: Formulation and application to ozone profile comparison, J. Geophys. Res., doi:10.1029/2005JD006122. Would this be a feasible approach for this study? For comparisons using reference data with a very high vertical resolution, such as ozone sondes, the transformation in Calisesi is not required because the ozone sonde's 'averaging kernel' peaks only near the measurement altitude (as it is a very localised measurement of the air it passes through). The comparisons discussed in the paper are not between two satellite retrievals, but rather between observed profiles (from MLS and ozone sondes) and co-located ozone analyses resulting from the assimilation experiments. For what I can see, the MLS data does not come with averaging kernels, so it is not possible to convolve the analyses with those in the comparisons. It is also noted that MLS, being a limb instrument, also has by design averaging kernels that strongly peak near the measurement altitude.

- P11 L29-31: Both instruments show reductions of the standard deviation (Fig 9). The one from O3-CCI/GOME-2 occurs over a wider vertical range than the one from O3-CCI/MIPAS, wheras the latter seems to have stronger localised reductions. Which of the two would be preferable for the assimilation as a whole and why? It is a tricky question to answer and deciding on which one of these two situations is preferable might depend on other considerations as well, for instance on the region where the improvement is found. They are both a sign of an improvement. In general the more one can reduce that standard deviation the better the fit to the independent data is. Here, the important point here is

not to say which one is better than the other but trying to identify where and how much the data from these two viewing geometries can be expected to make an impact.

**General question:**

The author demonstrates that the comparison with MLS and ozone sondes improves when GOME-2 and MIPAS based ozone profiles are assimilated, but if it is not too far out of scope of this paper, it would be interesting to get an indication of the change in the skill of the IFS in general as a result of the assimilation of the additional ozone input. E.g.: the effect on wind vectors or temperature.

This aspect was not considered in this work for two reasons. The first reason is that the focus was to exclusively look at the impact on ozone produced by the assimilation of limb and nadir data. The second reason is that in the current system, ozone is implemented as a univariate variable. This means that in general the assimilation of L2 ozone products has no impact on the rest of the system. In my experience, this is normally the case. In rare situations, however, a measurable impact can indirectly be triggered on some meteorological fields if the assimilation of the Level 2 (L2) ozone data is able to modify the data usage of the ozone-sensitive radiances (IR/O3) measured by instruments like IASI, AIRS, and CrIS. For this to happen, the L2 ozone product needs to have a sufficiently high number of observations and be able to modify the IR/O3 data assimilation in the UTLS (i.e. where these radiances have the highest sensitivity) and their data usage. If that occurs, a chain reaction can be triggered within the 4D-Var data assimilation. In an attempt to fit all data at once, 4D-Var can modify the assimilation of other high-impact observations (e.g. from sensors like the microwaves) that can, in turn, lead to a measurable impact on meteorological fields. However, this is not the norm, nor in my understanding is the assimilation of the IR/O3 radiances by all other Numerical Weather Prediction centres, thus adding that element in the paper could be misleading. For this reason, it is preferred not to discuss it.

**Typographical comments:**

- P2 L2: signature → ... the signing of an international treaty.
  Corrected as advised, thank you.
- P3 L5: greatly → very.
  Corrected as advised, thank you.
- P4 L28: **The** CCI ...
  Corrected as advised, thank you.
- P5 L10: verified? You may mean 'present'.
  The verb "verified" refers to the (LTE) conditions. I would prefer to keep the verb verify instead of using the verb "present".
- P5 L32: satellites (plural).
  Corrected as advised, thank you.
- P6 L16: "An example of **a** background error profile and **a** vertical correlation..."
  Corrected as advised, thank you.
- P6 L18: Introduction of acronym TCO3 without prior explanation (also not in table 1).
  The acronym TCO3 has been added to table 1. Thank you.
- P12 L24: IS equivalent.
  Since the sentence was formulated as a question, the auxiliary was already included at the start.
- P12 L25: what → why

The use of "why" in this case would not be correct. I really meant to ask what mechanism is or could be responsible for the improving the quality of the Exp/MIPAS ozone analyses at levels below the availability of the MIPAS observations. This was left unchanged.

- Figure 12: The plots are small and the black and blue are sometimes hard to distinguish with this line width. Would it be possible to provide larger plots with thicker line? One could try a 2-1-2 panel ordering instead of the current 3-2.
  The panels in figure 12 have been rearranged as asked. The line thickness has only marginally been increased as larger lines would have made the results for two experiments almost overlap when they do not.

  The initials were correct, but the "and" between the first and second co-authors was misspelled as "amd". Thank you for noticing it. It has now been corrected.

**Reply to Reviewer # 3:**

I thank Dr S. Chabrillat for his comments and suggestions, but it is noted that the location of specific text from the paper provided in his comments is always incorrect and that some of the minor comments asking for corrections on text were not find in the ACPD version of the manuscript, neither in my own local copy nor in the submitted version available on-line. I wonder if this review is based on the actual ACPD paper. I tried to address the points raised at the best of my ability all considering.

**General comments:**

I believe that this paper is a useful and valuable contribution to the field of ozone data assimilation but it fails to consider related work, and many appropriate references are missing. Hence while there is no need for any additional assimilation experiment, the text should still undergo major revisions.

I agree that many publications have focussed on ozone assimilation. The introduction refers to a number of studies performed with MIPAS (Dethof 2003, Wargan et al 2005, Geer et al 2006, Dragani 2013), MLS (Jackson 2007, Feng et al 2008), and MLS+OMI (Stajner et al 2008). I might not have covered the whole available literature but I do not agree it failed to consider related work. That said, I have now added a few more references.

1. This paper gives a false impression that ozone data assimilation is still in its infancy. There is a whole community working on this topic for a long time but none of its previous work is mentioned in the introduction nor considered in the discussion. I think that the introduction should be extended to provide proper context, and that this context should be used in the discussion.
   I agree with the Reviewer that there is a vast literature on ozone data assimilation, including some review papers, and as mentioned above the introduction has been extended. However, I do not agree that the paper lacks of context. As stated in the paper, the study started from two considerations: the first is the decision made by NASA in generating the MERRA2 reanalysis, in which the SBUV nadir profile assimilation was completely replaced by that of MLS (limb profiles) and OMI total columns. This contrasts with the normal trend in (NWP/reanalysis) data assimilation of using as many observations as possible. The second consideration was the lack of plans for additional limb instruments (neither on operational nor research platforms) in the foreseeable future. I believe this, as context, has been covered in the introduction and the discussion has been tailored to it.

2. The datasets assimilated in this study were developed for the ESA project O3-CCI. That project led to several validation papers which discuss extensively the uncertainties and information content of the corresponding datasets. Since a good evaluation of observational uncertainties is paramount for data assimilation, such prior work is highly relevant for this paper. Hence the O3-CCI validation papers should be at least cited in section 4, and the choices made for the present assimilation study should be discussed in this context.
   The revised version of the manuscript has been extended to include additional references from the existing literature.

**3.** The averaging kernels of GOME-2 nadir profiles are still not taken into account by IFS. This is a serious limitation of the present study, because many other assimilation systems now take properly into account such vertical smoothing errors. This limitation should at least be clearly stated in the conclusions and abstract of the paper: "This study demonstrates the potentials and limitations of each dataset and instrument type" – but only in the context of data assimilation with the current IFS at ECMWF.

Regarding the use of the averaging kernels (AKs) in the assimilation, the IFS version used here (e.g. the same run for the operational weather forecasts) does not include them at all (meaning they have not been implemented yet). Some preliminary tests have been performed, however, with the C-IFS version (now used as part of the Copernicus Atmosphere Monitoring Service). Results so far have shown at best a neutral impact of using the AKs in our ozone assimilation (A. Inness, personal communication), thus not really a limitation in practice. This has been commented in the paper.

**Specific comments:**

1. The introduction does not mention the results obtained in previous projects about data assimilation of stratospheric ozone, giving a false impression that ozone analyses are available only in two meteorological reanalyses (i.e. ERA-Interim and MERRA-2).

   This is not a reanalysis paper, thus I cannot justify a review of the ozone analyses in all available reanalysis, for which readers can refer to the literature (including information provided at www.reanalysis.org) and projects such as the SPARC Reanalysis Intercomparison Project (S-RIP). A comment has been added.

   Yet simultaneous assimilation of limb and nadir ozone datasets was reported and discussed as early as 2002 (Struthers et al., 2002). Nearly ten years ago, Lahoz et al. (2007) were already able to review this field. Even considering only the European projects, I believe that it is not possible to ignore such prior work as the ASSET intercomparison (Geer et al., 2006), the developments for the PROMOTE project (Viscardy et al., 2010) or the numerous results obtained for the MACC series of projects (see e.g. Inness et al., 2013; Inness et al., 2015).

   I agree that some of these papers discussed assimilation of nadir and limb data in various combinations, and a number of studies were referred to in the paper already. The introduction has been further extended as appropriate, but note my remark below and in point 5 regarding the MACC IFS ozone, as well as the fact that the MIPAS assimilation in IFS discussed in Geer et al, 2006 is partly based on the Dethof (2003) work cited in the paper.

   The absence of any citations about ozone assimilation in MACC is especially strange, because the MACC projects were coordinated by the same Institution as the author (ECMWF) and relied on a version of the same model (IFS). MACC allowed an intercomparison of the ozone analyses delivered in Near Real Time by four different systems assimilating nadir and/or limb datasets (Lefever et al., 2015). Even though the assimilation experiments were very different, this earlier study reached a very similar conclusion with a very similar Data Assimilation System (DAS): "IFS-MOZART is able to deliver realistic analyses of ozone both in the troposphere and in the stratosphere, but this requires the assimilation of observations from nadir-looking instruments as well as the assimilation of profiles, which are well resolved vertically and extend into the lowermost stratosphere".

   I intentionally avoided to mention the MACC system. The similarities between the MACC and operational IFS are related to the meteorological part. There are a number of differences between

the two ozone analysis systems and not always a result obtained with one system can be extended to the other (e.g. the assimilation of the ozone-sensitive radiances was found beneficial in IFS and operationally assimilated since November 2011 but to produce neutral to slightly negative impact in the MACC-IFS system - where they are not assimilated yet; on the other hand the MACC system uses MLS, which was found to degrade the ozone analyses in the UTLS region when tested with the operational IFS – please see also my reply to point 5 below). The Lefever et al. (2015) paper has now been referred to.

Overall it is necessary to extend significantly the introduction in order to provide the missing context, and to take prior work into consideration in the discussion of the results (section 5.2.2). Again, I suspect the Reviewer is not referring to the current version of the paper. That said and as mentioned above, the introduction has been extended as appropriate, see also my reply throughout this document.

2. P.1, line 24: the concern for the ozone decline is primarily due to the expected increase of Ultraviolet radiation at the surface. This should be mentioned in the introduction, along with a general reference about the issue.
An explicit sentence has been added to the current paper.

3. The description of the assimilated datasets (section 2) and the data quality analysis (section 4) both fail to consider the extensive validation work realized for the O3-CCI ozone datasets. At least three papers investigate the quality of the MIPAS and GOME-2 datasets which are assimilated here. Hassler et al. (2014) present an overview of stratospheric ozone profile measurement data, document measurement techniques, spatial and temporal coverage, vertical resolution, native units and measurement uncertainties; Laeng et al. (2015) and Keppens et al. (2015) compared the available retrieval algorithms for MIPAS and GOME-2, respectively, explaining the choice of the algorithms selected for the O3-CCI datasets. These studies about observational uncertainties should be used in the description of the assimilated datasets and could be useful for the discussion of the results. Miles et al. (2015) should be cited, not only as a reference for the assimilated GOME-2 dataset, but also for its specific validation results.
As mentioned above, the paper has been extended as appropriate.

4. The limited vertical resolution of the GOME-2 dataset should be explained more extensively, citing a specific paper (e.g. Keppens et al., 2015) in addition to the overarching reference (Rodgers, 2000). Since GOME-2 profiles have "between 5 and 6 degrees of freedom", figure 2 does not show their vertical resolution. It shows instead the vertical grid of the retrieved product. This confusion could be seriously misleading for the novice reader. While it is less of a concern thanks to its limb-viewing geometry, MIPAS does not have perfect vertical resolution either (von Clarmann and Grabowski, 2006; Laeng et al., 2015). This should also be mentioned in section 2.
Correct, figure 2 shows the product vertical grid. As matter of fact the legend states: "*Schematic of the vertical coverage and vertical resolution provided by the CCI GOME-2 (red lines) and the CCI MIPAS (blue lines) retrieval algorithms*". I have rephrased the original legend as follows: "*Schematic of the two ozone products' vertical coverage and vertical resolution as provided*….". That said, as also pointed out by Reviewer #2, the main text could be misleading, and it has been changed from "*… the two instruments used in the present study offered different horizontal (Figure*

*1) and vertical (Figure 2) coverages*" to "*...the two sets of retrievals used in the present study offered different horizontal (Figure 1) and vertical (Figure 2) coverages*".

5. Description of the DAS (section 3): what is the IFS version number ("cycle") used here? How does it compare with the versions used in ERA-Interim (Dragani, 2011) and the MACC reanalysis (Inness et al., 2013) as far as ozone assimilation is concerned?

   ERA-Interim was run with a 2006 IFS version (CY31R2), the current experiments with a much newer version CY40R1 (2013), the MACC reanalysis with cycle CY36R4 (2010). The experiments here benefit from higher horizontal and vertical resolutions than the two analyses. In addition and specifically on ozone, there are many differences in terms of assimilated ozone data, ozone bias correction and associated anchor, ozone data quality control, and forecast model/chemistry. Because of these differences, a one-to-one comparison between the three corresponding ozone analyses is not immediate. These have now been added to the paper in the main text and a new table 3.

6. The modelling of ozone in IFS is not properly described, again leading to a lack of context for the discussion of the results. How is ozone photochemistry represented in the forecast model? Assuming that the parameterization by Cariolle and Teyssèdre (2007) is used here, this is not an explicit modelling of ozone photochemistry. So what does the sentence (p.5, line 15) "In this forecast model and analysis system, ozone is fully integrated (Dethof and Hólm, 2004)" mean exactly? The parameterization by Cariolle and Teyssèdre has some limitations which should be stated as they could explain some of the assimilation results.

   As said in the paper, the parametrization follows the Cariolle and Déqué (1986) [CD86] for the homogeneous chemistry. An additional term to parametrize the heterogeneous chemistry was added to the original CD86 formulation, as discussed in Dethof and Holm (2004). The scheme has been used in the same formulation since. However, the coefficients of this linear regression are updated regularly (typically every two years). These are provided by Daniel Cariolle and collaborators. The above mentioned Cariolle and Teyssèdre (2007) paper describes how they are calculated. Regarding the limitation of this scheme, I do agree it is not perfect, for example the coefficients are produced with a 2D model that does not include explicitly the heterogeneous chemistry, reason why an additional term had to be added in the IFS. Work is on-going to address these points, and preliminary results are encouraging (though on the medium and long forecast range). A comment has been added.

   The expression "ozone is fully integrated" means that ozone in IFS is a prognostic variable like, for instance, temperature, and not a climatology. The sentence has been changed from "fully integrated" to "prognostic variable".

7. P. 5, lines 23 -26: "accounting for the vertical sensitivity of any retrieved product as provided by the data averaging kernels (AKs) is currently not possible in the IFS". Please provide a reference for this limitation of IFS. Many other DAS now do take AKs into account, implementing a straightforward approach (explained e.g. by von Clarmann and Grabowski, 2006). Hence this limitation of IFS is a key caveat for this study because it limits the applicability of its findings to other DAS (see third major comment).

As mentioned above, I believe the review is not based on the current ACPD paper, as matter of fact P. 5, lines 23 -26 of the manuscript under review does not refer to the mentioned sentence, which is instead at p. 6, lines 12-15.

Regarding the specific point (see also my reply to point 3 above), in IFS the use of the AKs has not yet been implemented, thus using them is not an option. In the MACC/CAMS IFS (please note this is a different system from the operational IFS) some tests have been performed. However, the results obtained when assimilating the ozone data with the AKs compared to the same assimilation without AKs are at best neutral (A. Inness, personal communication). A comment has been added to the paper.

"With such an approximation, the vertical spread of the ozone information provided by the assimilated ozone observations depends on the background error variances and covariance (B) for ozone". Please provide a reference about this.

By design, the location where 4D-Var increments are placed, i.e. where an observation has the largest impact on the analysis, depends on both the background error and where the data show sensitivity, i.e. the AKs in the case of retrievals. A good proxy is represented by the maximum of the convolution of the background error and AKs. With the box-car approximation, each AK function is assumed to be 1 over the layer it refers to and 0 otherwise, thus the impact of the data only depends on the background error and localized where this is maximum. Han and McNally (2010) showed an illustration of it in the case of IASI radiance assimilation, using the Jacobians in place of the AKs. This part has been explained more plainly.

This approximation also fails to properly take vertical smoothing errors into account, and may constrain the analysis with a priori information contained in the retrievals. For example, in some viewing geometries the GOME-2 retrievals do not contain any usable information close to the surface.

Correct. As it is, the system assimilates the level 2 product consisting of the information provided by the measurements as such, and by the a priori used during the retrieval scheme, noting that:

- The a priori is, in general, information and as such there is nothing wrong with assimilating a product that includes it as long as that a priori does not misrepresent the ozone state variability. (I would also argue that if the information in a retrieval - whether from the measurements or the a priori – is not usable, this should also be reflected in the observation uncertainty.)

- The assimilation also relies on the model background, on the physical consistency with other variables (temperature, winds, etc…), and in complex systems - like those used in NWP – also on the simultaneous assimilation of other observations. Different assimilation techniques, e.g. KF-based, could be more sensitive to the way the observations are used, thus allowing one to better exploit these additional pieces of information than 4D-Var does. In the KF, for instance, the model is integrated forward in time, and as a new observation is available it is used to reinitialize the model before continuing the integration. 4D-Var is an initial condition problem, the aim is to find the initial condition that gives a trajectory that best fits all observations in the assimilation window ("all" here means all data used to constrain any variable included in the state vector) and the model background. For that reason, demonstrating that the assimilation of 'purged' retrievals in a 4D-Var NWP system is better than that of conventional retrievals can be very difficult (see also my comments on the use of the AKs).

8. Figures 5-6 could be quite interesting for the retrieval and validation communities who are not familiar with the estimation of observational errors allowed by data assimilation (i.e. the method by Desroziers et al., 2005). The attempt to explain this method (p.6, lines 28 - 32) is quite unclear, it should be re-written and expanded.

   The text has been rephrased. However, the reader should refer to the literature for more detailed explanation.

   P. 7, lines 18 -19: "the differences between provided and estimated uncertainties appear to be rather large". This is an important result for the aforementioned communities (even though these uncertainties "only represent up to about 4% of the observation values"). Hence it should be shown, i.e. figures 5-6 should be expanded with similar latitude - pressure cross-sections showing the provided uncertainties and using the same color scale.

   The figures have been modified to include also the differences relative to the observations, as advised. The text has also been modified accordingly.

9. P. 7, lines 1-4: paragraph is unclear, please re-write. "The reason for this is still under Investigation at the time of writing": indeed, this is not expected from the comparisons of ozone total columns between SBUV and GOME (Chiou et al., 2014)

   Please note the figure shows the differences between GOME-2 and the control analyses (co-located to the observations). These analyses are yes constrained by the SBUV retrievals, but also By the SCIAMACHY TCO3, in addition to IR/O3 radiances, and affected by the way the assimilation is performed (e.g. GOME-2 is assimilated as a profile, SBUV as a 6-layer partial column product) and by the model background. It is also noted that the v8 NRT SBUV product is assimilated here while the Chiou et al paper is based on the v8.6 reprocessed SBUV profiles. Without assessing the impact of each of these elements individually, one can only speculate on the element responsible for those differences. This point has been highlighted in section 4 of the paper.

10. P. 7, lines 9-11: "The first-guess check implemented in the IFS discards all observations that, after successfully pass the data quality control, show discrepancies from the background of 30DU or more over the column". Please re-write (e.g. "after having successfully passed").

    Actually the sentence in the manuscript under review reads as *"…, after they successfully pass the …"*. No change has been made to the paper.

    "Figure 4 shows that the observations from both instruments are well within such a threshold". Figure 4 cannot be used to justify this Background Quality Check (BGQC) because it shows a global mean of the O-A departures while the BGQC is applied to individual observations.

    In the paper under review, the sentence reads "*Figure 4 shows that on average the observations from both instruments are well within such a threshold, although it is noted that individual observations might have shown residuals from the background larger than 30DU*". No change has been made to the paper.

**Minor Comments:**

- P. 3, line 23: remove extra closing parenthesis.
  I could not find any extra closing parenthesis in the manuscript under review.
- Legend of figure 1 : state the year plotted here (i.e. 2008).

  The year has been added to the legend.

- Figure 3: right panel would be much nicer as a color-coded contour plot.

  This plot (as stated) is from Dragani and McNally (2013). I appreciate the suggestion, but unfortunately I no longer have the original data to reproduce the plot in colours.

- P. 7, line 6: "instruements" → "instruments"

  I could not find this typo in the version under review.

- P. 7, line 9: "that, after successfully pass the data quality control" – please re-write
  In the ACPD manuscript under review, the sentence actually reads "*The first-guess check implemented in the IFS discards all observations that, after they successfully pass the data quality control, show…*".

- P. 7, line 18: "differencies" → "differences"

  I could not find this typo in the version under review.

- Table 1: Acronyms "BUV" and "ODS" are missing. I suggest to list first the satellite instruments, followed by the other acronyms

  ODS is defined in the main text (page 2 line 3). BUV, which is first mentioned in page 2 line 13, is instead defined in table 1 as advised in page 2 line 4 "*acronyms not defined in the text can be found in table 1*". Table 1, as just said, collects all the acronyms that, if defined in the text, could have made the sentences too long. Considering that new acronyms were added during the review process, table 1 was split into two tables, one for satellite platforms and instruments, and the other for all other acronyms.

- Figure 7: color scale does not work well. Use same red-blue scale as for figures 5 and 6.
  The colour scale from figure 7 onwards had to be changed because when we tried to define 10 colour variations for the positive values from the light pink to the deep red, the differences between some of the shades were so little that made it difficult to distinguish them in the plot. As a consequence we had to change the pink-red positive part of the colour scale to use a yellow-orange-red one.

[revised manuscript text omitted]